# LaMg$_6$Ga$_6$S$_{16}$: a chemical stable divalent lanthanide chalcogenide

Yujie Zhang[1], Jiale Chen[1], Kaixuan Li[1], Hongping Wu[1], Zhanggui Hu[1], Jiyang Wang[1], Yicheng Wu[1] & Hongwei Yu ●[1] ✉

Divalent lanthanide inorganic compounds can exhibit unique electronic configurations and physicochemical properties, yet their synthesis remains a great challenge because of the weak chemical stability. To the best of our knowledge, although several lanthanide monoxides epitaxial thin films have been reported, there is no chemically stable crystalline divalent lanthanide chalcogenide synthesized up to now. Herein, by using octahedra coupling tetrahedra single/double chains to construct an octahedral crystal field, we synthesized the stable crystalline La(II)-chalcogenide, LaMg$_6$Ga$_6$S$_{16}$. The nature of the divalent La$^{2+}$ cations can be identified by X-ray photoelectron spectroscopy, X-ray absorption near-edge structure and electron paramagnetic resonance, while the stability is confirmed by the differential thermal scanning, in-situ variable-temperature powder X-ray diffraction and a series of solid-state reactions. Owing to the particular electronic characteristics of La$^{2+}$(5d$^1$), LaMg$_6$Ga$_6$S$_{16}$ displays an ultrabroad-band green emission at 500 nm, which is the inaugural instance of La(II)-based compounds demonstrating luminescent properties. Furthermore, as LaMg$_6$Ga$_6$S$_{16}$ crystallizes in the non-centrosymmetric space group, $P\overline{6}$, it is the second-harmonic generation (SHG) active, possessing a comparable SHG response with classical AgGaS$_2$. In consideration of its wider band gap ($E_g$ = 3.0 eV) and higher laser-induced damage threshold (5×AgGaS$_2$), LaMg$_6$Ga$_6$S$_{16}$ is also a promising nonlinear optical material.

Lanthanide inorganic compounds with low oxidation state (+2) that are capable of exhibiting intriguing physicochemical properties due to the presence of outer shell 4$f$ or 5$d$ conduction carriers in divalent lanthanides ions have showcased the immense potential for application in various frontier fields such as superconductivity, magnetics, photoluminescence[1–8]. However, one intractable drawback to divalent lanthanide compounds is the chemical stability, which seriously precludes their development[9,10]. Recently, although several new types of divalent lanthanide monoxides epitaxial thin films, including YO and LaO, have been prepared, the surfaces of these films must be capped in-situ AlO$_x$ layer to prevent the oxidation at room temperature[9,11]. Thus, synthesizing the chemically stable divalent lanthanide

compounds is still faced by great challenges. To date, as we know, no any successful stable crystalline divalent lanthanide chalcogenide has been synthesized.

Based on the first-principles calculations, *Li* et al. have uncovered the octahedral crystal field is vitally pivotal for the formation of the divalent lanthanum in LaO[12]. We have also noticed that almost all the divalent lanthanides (Ln$^{2+}$) in lanthanide monoxides and monochalcogenides are coordinated by six Q (Q = O or S) atoms to form the [LnQ$_6$] octahedra[4,13]. On the contrary, the high-oxidation-state lanthanides (Ln$^{3+}$) is typically found in the high-coordinated [LnQ$_x$] (x = 7 or 8) polyhedra, e.g., La$_2$S$_3$[14], LaGaS$_3$[15], La$_2$Ga$_2$GeS$_8$[16], La$_6$MgGe$_2$S$_{14}$[17], K$_3$LaP$_2$S$_8$[18], KYGeS$_4$[19,20], Ba$_3$La$_4$O$_4$(BO$_3$)$_3$X (X = F, Cl, Br)[21], which also

[1]Tianjin Key Laboratory of Functional Crystal Materials, Institute of Functional Crystal, College of Materials Science and Engineering, Tianjin University of Technology, Tianjin, China. ✉e-mail: yuhw@email.tjut.edu.cn

conforms to Pauling's well-known second rule, i.e., high-valence is favored for high coordination[22]. On the other hand, *Evans* and *Meyer et al.'s* research show that the construction of the proper anionic frameworks in combination with lanthanide cations can enhance chemical stability by its gain in lattice energy[23], as corroborated via the synthesis of a series of stable divalent lanthanide organic complexes, including [(18-crown-6)K][(C5H4SiMe3)3Y][24], [K(18-crown-6)(OEt2)] [(C5H3(SiMe3)2–1,3)3La][10], and [K([2.2.2]crypt)][LaCp″3](Cp″=1,3-(SiMe3)2C5H3), [2.2.2]crypt=4,7,13,16,21,24-hexaoxa-1,10-diazabicy-clo[8.8.8]hexacosane)[10].

Clearly, the above studies have implied that the strong octahedral crystal fields and proper anionic framework are crucial for the formation of stable divalent lanthanide compounds. In the recent research, by adopting octahedra to couple tetrahedra single/double chains strategy, a stable crystalline [Mg/Ga-S]∞ anionic framework with octahedral channel ($C_{3h}$) has been constructed by our group and *Pan's* group[25,26], where alkali and alkaline-earth or even other monovalent or divalent cations can be filled (Fig. 1), and all of the resulting compounds exhibit the similar crystal structures, *e.g.*, AMg3M3Q8 (A=Li, Na, Ag; M=Al, Ga; Q = S, Se) and AeMg6Ga6S16 (Ae=Ca, Sr, Ba). Based on these, we speculated that the framework should be also available for the syntheses of the divalent lanthanide chalcogenide because of its particular structural feature and strong accommodating ability for a wide range of elements and oxidation states. Guided by these ideas, we introduced the lanthanum (La) into the stable [Mg/Ga-S]∞ anionic framework and successfully synthesized the crystalline La(II)-chalcogenide, LaMg6Ga6S16. In its structure, the stable [Mg/Ga-S]∞ framework channels create the strong [LaS6] octahedra crystal field, which results in the formation of stable divalent $La^{2+}$ possessing the presence of outer shell $5d^1$ conduction carriers. Interestingly, owing to the unique electronic characteristics of $La^{2+}(5d^1)$, LaMg6Ga6S16 exhibits an ultrabroad-band green emission at 500 nm with an excitation of 360 nm. This is the inaugural instance of La(II)-based compounds to display luminescent properties. Additionally, as LaMg6Ga6S16 crystallizes in the noncentrosymmetric space group of *P*−6, the excellent nonlinear optical (NLO) properties are also observed in LaMg6Ga6S16, including the relatively large second-order harmonic generation (SHG) response (~0.8×AgGaS2), wide band gap ($E_g$ = 3.0 eV), high laser-induced damage threshold (LIDT) (5 × AgGaS2), and wide transparent window (0.41-20 μm). These make LaMg6Ga6S16 a promising NLO

crystal. Herein, we will report its synthesis, structure, and luminescent and NLO properties.

## Results and discussion

### Experimental synthesis and structure determination of LaMg6Ga6S16

Polycrystalline LaMg6Ga6S16 was synthesized through a conventional solid-state technique in sealed silica tubes at 1233 K and the purity of phase was verified by the powder X-ray diffraction (XRD) (Supplementary Fig. 1). Furthermore, the energy-dispersive spectroscopy measurement showed the existence of La/Mg/Ga/S, and their average atomic ratios were approximately equal to the theoretical ones, 3.45%, 20.69%, 20.69%, and 55.12% (Supplementary Fig. 2). Then, the millimeter-sized single crystals of LaMg6Ga6S16 were grown by melting and re-crystallizing the stoichiometric pure phase. By using these crystals, the crystal structure of LaMg6Ga6S16 was determined by single crystal XRD. It indicates that LaMg6Ga6S16 crystallizes in the noncentrosymmetric hexagonal space group *P*-6 (*No.*174), with cell parameters of *a* = 16.7154(5) Å, *c* = 7.4147(3) Å, and *V* = 1794.15(13) Å³ (Supplementary Table 1). In the asymmetric unit, there are three unique La, six unique Mg, three unique Ga, and eleven S atoms (Supplementary Table 2). The Mg atoms are six-coordinated forming [MgS6] octahedra with the Mg−S distances ranging from 2.482(11) to 2.834(18) Å. All of the Ga atoms are coordinated by four S atoms to form [GaS4] tetrahedra, and the Ga-S distances range from 2.226(7) to 2.333(6) Å. The La atoms are coordinated by six S atoms to form [LaS6] octahedra with La−S distances ranging from 2.963(7) to 2.994(7) Å. All of these distances (Supplementary Table 3) are consistent with those in other chalcogenides[17,27,28].

The structure of LaMg6Ga6S16 is shown in Fig. 2. Clearly, LaMg6Ga6S16 features a three-dimensional (3D) framework with $C_{3h}$ symmetry along the *c* axis and constructed by the [MgS6] octahedra coupling [GaS4] tetrahedra single/double chains (Fig. 2a). In detail, the MgS6 octahedra are connected with each other via corner-sharing and face-sharing (in the *a-b* plane) and edge-sharing (along the *c*-axis) to fabricate a [Mg-S]∞ framework, as shown in Fig. 2b. While the [GaS4] tetrahedra are connected via corner-sharing to form two types of Ga-S chains along the *c*-axis, *i.e.*, [Ga(1)S3]∞ single chains (Fig. 2c) and [Ga(2,3)2S4]∞ double chains (Fig. 2d). Furthermore, the resulting [Ga(1)S3]∞ single chains are connected and fixed in the [Mg-S]∞ framework by the Ga-S bonds to create the [Mg/Ga-S]∞ framework (Fig. 2e), which are further linked by the [Ga(2,3)2S4]∞ double chains to construct the 3D framework structure of LaMg6Ga6S16. The La atoms fill the channel-like cavities of the 3D framework to balance the residual charges (Fig. 2f).

Interestingly, the La atoms in LaMg6Ga6S16 exhibit the scarcely seen divalent state (+2), which was only reported in three metastable inorganic compounds LaO, LaS and LaS2 with multiple phase transitions (α: *P21/b*; β: *Pnam*; γ: *P4/nmm*)[29,30] and two organic complexes [K(18-crown-6)(OEt2)][(C5H3(SiMe3)2−1,3)3La] and [K([2.2.2]crypt)] [LaCp″3](Cp″ = 1,3-(SiMe3)2C5H3), [2.2.2]crypt=4,7,13,16,21,24-hexaoxa-1,10-diazabicyclo[8.8.8]hexacosane)[4,9,10]. To ascertain the oxidation state of $La^{2+}$ in LaMg6Ga6S16, X-ray photoelectron spectroscopy (XPS) measurement and analysis for La metal, LaMg6Ga6S16, and La2S3 were conducted and demonstrated, as shown in Fig. 3a. The results show that the peak position of La $3d_{3/2}$ (851.5 eV) and $3d_{5/2}$ (834.1 eV) in LaMg6Ga6S16 is located between those of La $3d_{3/2}$ (851.7 eV) and $3d_{5/2}$ (834.9 eV) in La metal ($La^0$) and La $3d_{3/2}$ (851.2 eV) and $3d_{5/2}$ (833.7 eV) in La2S3 ($La^{3+}$), suggesting the divalent state (+2) of La in LaMg6Ga6S16[9]. To better characterize the chemical valence of La in LaMg6Ga6S16, the synchrotron X-ray absorption spectroscopy (XAS) measurements of LaMg6Ga6S16 and La2S3 were performed. As indicated by the La *L*-edge X-ray absorption near-edge structure (XANES) spectra (Fig. 3b), LaMg6Ga6S16 exhibits an absorption edge with energy lower than that of La2S3, indicating a lower valence state La in LaMg6Ga6S16 than that of La2S3 (+3)[31,32]. This is in good agreement with the XPS results. Further,

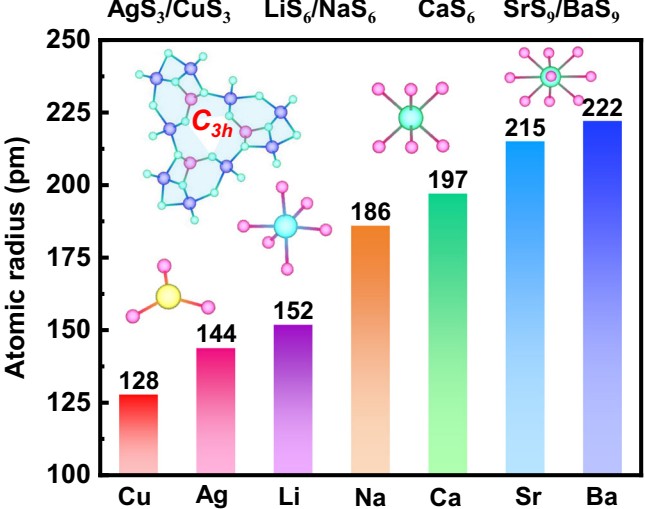

**Fig. 1 | The accommodating ability of [Mg/Ga-S]∞ framework.** The accommodating ability of [Mg/Ga-S]∞ framework. Statistics on radius and the coordination of a range of atoms filled in [Mg/Ga-S]∞ framework. This framework can be filled with alkali and alkaline-earth or even other monovalent divalent cations.

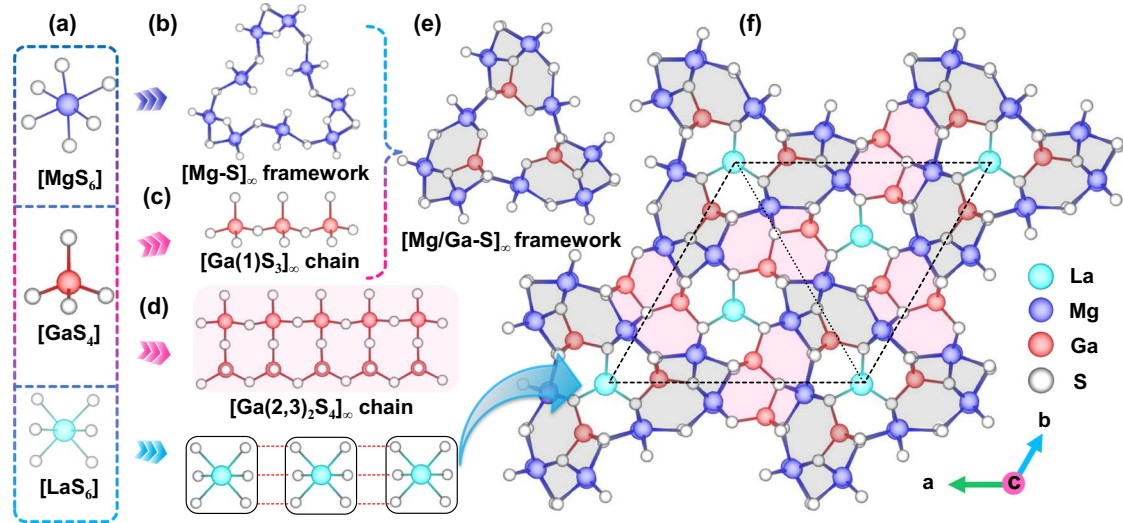

**Fig. 2 | Crystal structural features of LaMg₆Ga₆S₁₆.** MgS₆ octahedron, GaS₄ tetrahedron, and LaS₆ octahedron **a**; [Mg-S]∞ framework **b**; 1D [Ga(1)S₃]∞ chain **c**; [Ga(2,3)₂S₄]∞ chain **d**; [Mg/Ga-S]∞ framework **e** and structure of LaMg₆Ga₆S₁₆ viewed along the *c*-axis, the dashed line represent single unit cell **f**. The MgS₆ octahedra firstly connect with each other via corner-sharing (in the *a-b* plane) and edge-sharing (along the *c*-axis) to form Mg-S framework with [Ga(1)S₃]∞ single chains connected and fixed in the framework by the Ga-S bonds. Then, these adjacent open frameworks are further linked by the [Ga(2,3)₂S₄]∞ double chains to create the [Mg/Ga-S]∞ framework. The color codes for the atoms are blue: La, violet: Mg, red: Ga, grey: S.

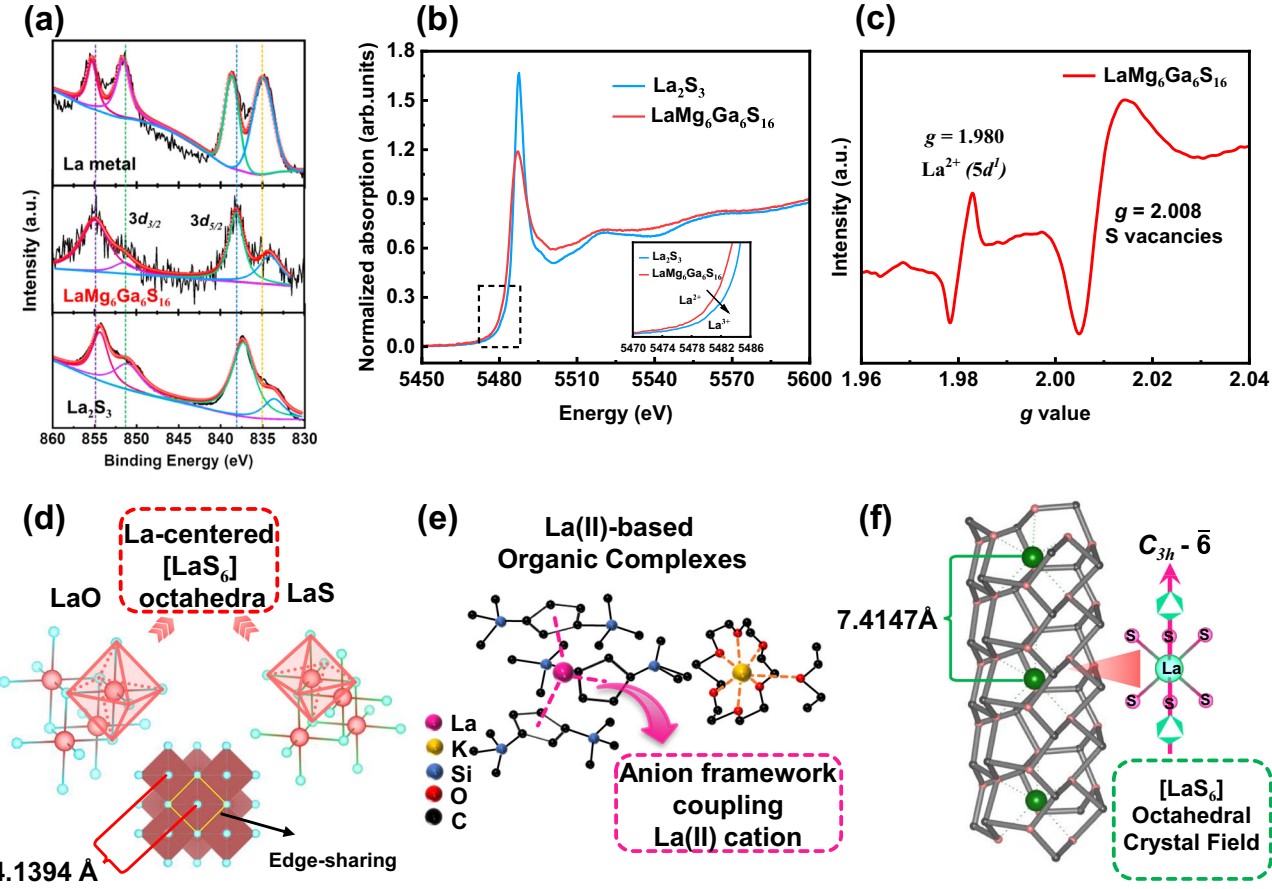

**Fig. 3 | Identification of La(II) valence states and structural analysis in LaMg₆Ga₆S₁₆.** La 3*d* XPS spectrum with fitting curves for the La metal powder, LaMg₆Ga₆S₁₆, and La₂S₃ **a**; La *L*-edge normalized XANES spectra of LaMg₆Ga₆S₁₆ (red line) and La₂S₃ (blue line); Inset (from black dashed square region) gives XANES spectra between 5470 eV and 5486 eV of LaMg₆Ga₆S₁₆ (red line) and La₂S₃ (blue line), the black arrow represents the increase in energy from low to high **b**; EPR spectrum of LaMg₆Ga₆S₁₆ **c**; Crystal structure of La(II)-based inorganic compounds: LaO and LaS **d**, the color codes for the atoms are red: La, blue: O, green: S; Crystal structure of La(II)-based organic complexes: [K(18-crown-6)(OEt₂)] [(C₅H₃(SiMe₃)₂–1,3)₃La] **e**, the pink arrow indicates the coupling between anion framework and La(II) cation, and the color codes for the atoms are pink: La, yellow: K, blue: Si, red: O, black: C; [Mg/Ga-S]∞ framework and the LaS₆ octahedron **f**, red triangle indicates the coordination environment of the La atoms, and the color codes for the atoms are green: La, black: Mg, pink: S.

we also used electron paramagnetic resonance (EPR) to characterize the La$^{2+}$ (5$d^1$) in LaMg$_6$Ga$_6$S$_{16}$ sample. As shown in Fig. 3c, a distinct EPR signal is observed for LaMg$_6$Ga$_6$S$_{16}$ at $g = 1.980$, which could be attributed to an unpaired electron interacting strongly with the nucleus of $^{139}$La[23]. The similar EPR signals have been well-reported on defective [K(18-crown-6)(OEt$_2$)][(C$_5$H$_3$(SiMe$_3$)$_2$–1,3)$_3$La] and [K([2.2.2] crypt)][LaCp″3] and can be considered as the signature of the existence of La$^{2+}$ [10]. In addition, the bond valence sums calculations result in the values of 1.96–2.13 for La$^{2+}$, 1.80–2.05 for Mg$^{2+}$, 2.97–3.00 for Ga$^{3+}$, and 1.82–2.18 for S$^{2-}$ [33]. All of these indicate the nature of the divalent La$^{2+}$ cations in LaMg$_6$Ga$_6$S$_{16}$.

Further, the thermal behavior of LaMg$_6$Ga$_6$S$_{16}$ was studied by differential thermal scanning (DSC) measurements. Clearly, only one endothermic peak at 1140 °C was observed on the heating DSC curve (Supplementary Fig. 3), suggesting that LaMg$_6$Ga$_6$S$_{16}$ did not undergo the decomposition and structural phase transitions when the temperature was increased from room temperature to 1140 °C. Moreover, in-situ variable-temperature powder X-ray diffraction and a series of solid-state reactions in the sealed silica tubes with the different calcinated temperatures show LaMg$_6$Ga$_6$S$_{16}$ has no phase transition when its polycrystalline sample was heated from 10 K to 1273 K (Supplementary Fig. 4), which also manifest that LaMg$_6$Ga$_6$S$_{16}$ is thermally stable. Meanwhile, the crystal of LaMg$_6$Ga$_6$S$_{16}$ was placed in the air and water at room temperature for one week with no decomposition or degradation observed (Supplementary Fig. 5). In addition, Global Instability Index (GII) of LaMg$_6$Ga$_6$S$_{16}$ is calculated[33,34], and the result (0.088) is lower than 0.2 $v.u.$ That also indicates the structural stability of LaMg$_6$Ga$_6$S$_{16}$ [35–37].

The nature of the stable divalent La$^{2+}$ cations in LaMg$_6$Ga$_6$S$_{16}$ could be attributed to the unique [Mg/Ga-S]$_\infty$ anionic framework. Comparing LaMg$_6$Ga$_6$S$_{16}$ with LaO and LaS, it can be seen that these La$^{2+}$ cations exhibit the similar coordination features, $i.e.$, La$^{2+}$ cations are six-coordinated in LaO$_6$ or LaS$_6$ octahedral crystal fields (Fig. 3d), while the previous studies in LaO also elucidate that octahedral crystal field is helpful for the formation of divalent lanthanum[12]. In LaO and LaS, LaO$_6$ and LaS$_6$ octahedra are connected with each other via edge-sharing to build the whole structure, respectively. According to Pauling's third rule[22], such connections are disadvantageous for structural stability owing to the increased cation–cation electrostatic repulsion. Also, the calculated results of their GII show that LaO (0.215) and LaS (0.349) have greater than 0.2 valence unit ($v.u$)[34], which indicates their structures are indeed metastable. Importantly, previous research finds that constructing the proper anionic frameworks to couple lanthanum cations can enhance the chemical stability of compounds by effectively harnessing the gain in lattice energy[23]. The formation of both divalent lanthanum organic complexes is an excellent example to demonstrate this concept (Fig. 3e). Similar to the case with organic complexes, we developed a crystalline [Mg/Ga-S]$_\infty$ structural framework by adopting octahedra coupling tetrahedra single/double chains strategy. Of note, such a framework possesses interconnected structures in which the neatly arranged [Ga-S] chains were connected by the [Mg-S] framework via covalent bonds. Such interconnected structure endowed [Mg/Ga-S]$_\infty$ structural framework with strong chemical stability, which has also been demonstrated in the reported covalent organic frameworks[38]. Additionally, the stable structural framework can be used as a template to accommodate a series of A atoms (A=Li, Na, Ca, Sr, Ba, and even La) while spatially confining these atoms into atomic-scale channels via coordination configurations. In particular, the coordination bond lengths of A-S are in the range of 2.934–3.138 Å, which provides a suitable micro-environment for La because the bond lengths of La-S are about 3.000 Å[16,39]. More importantly, the crystalline [Mg/Ga-S]$_\infty$ framework channels possess $C_{3h}$ symmetry along the $c$-axis, in which the six-coordinated LaS$_6$ octahedral crystal field can be created (Fig. 3f). That facilitates the formation of divalent lanthanum when introducing La into the crystalline [Mg/Ga-S]$_\infty$ anionic

framework. Further, the LaS$_6$ octahedra in LaMg$_6$Ga$_6$S$_{16}$ are isolated and aligned arrangements along the $c$-axis with a longer La-La distance of 7.4147 Å than 4.1394 Å in LaS, which greatly reduces electrostatic repulsion between La$^{2+}$. These structural attributes of LaMg$_6$Ga$_6$S$_{16}$ will be able to promote it exhibiting good chemical stability.

**Photoluminescence (PL) properties**

Given the electronic characteristics of divalent lanthanum, we investigated the luminescence features of LaMg$_6$Ga$_6$S$_{16}$. The PL excitations at room temperature (298 K) were measured under the excitation of 340–380 nm. As shown in Fig. 4a, the optimal excitation wavelength is about 360 nm. Under the excitation of 360 nm at room temperature, LaMg$_6$Ga$_6$S$_{16}$ shows an ultrabroad-band green emission at 500 nm with a full width at half maximum (FWHM) of 127 nm (Fig. 4b). The ultrabroad emission band cover almost the whole visible light region and could find applications in the field of human-centric full-visible-spectrum lighting[40]. Meanwhile, this characteristic green emission under 360 nm excitation endows LaMg$_6$Ga$_6$S$_{16}$ with the potential light-emitting diode application under the excitation of commercial near ultraviolet chips[41]. Further, to investigate the origin of luminescence properties, the thermoluminescence (TL) measurement of LaMg$_6$Ga$_6$S$_{16}$ is performed. As shown in Fig. 4c, the sample shows a very weak TL glow curve in the range of 290 K to 450 K, indicating a low content of defects in LaMg$_6$Ga$_6$S$_{16}$. By fitting the TL curve with two Gaussian bands peaking at 346 K and 384 K, the characteristic trap depths ($E_T$) were estimated to be 0.69 and 0.77 eV by using the crude relationship $E_T = T_m/500$ eV, where $T_m$ represents the temperature (K) of the TL fitting peak[42,43]. In view of the EPR results ($g = 2.008$) (Fig. 3c), it is evident that the two trap depths originate from the intrinsic defects, corresponding to the slight S vacancies defect[44,45]. Based on these studies, the strong green emission observed in LaMg$_6$Ga$_6$S$_{16}$ does not stem from intrinsic defects of exceedingly low content. In order to find out the origin of PL property of LaMg$_6$Ga$_6$S$_{16}$, we also measured the luminescence features of CaMg$_6$Ga$_6$S$_{16}$ and SrMg$_6$Ga$_6$S$_{16}$, which are isomorphous to LaMg$_6$Ga$_6$S$_{16}$ with chemical substitutions from La to Ca or Sr. Experimental results indicate CaMg$_6$Ga$_6$S$_{16}$ (Fig. 4d) and SrMg$_6$Ga$_6$S$_{16}$ (Supplementary Fig. 6) have no PL emission. Also, when the polycrystalline samples of CaMg$_6$Ga$_6$S$_{16}$, SrMg$_6$Ga$_6$S$_{16}$ and LaMg$_6$Ga$_6$S$_{16}$ were radiated by the UV irradiation, only LaMg$_6$Ga$_6$S$_{16}$ exhibited the green light emission (Supplementary Fig. 7). These results suggest that the PL property of LaMg$_6$Ga$_6$S$_{16}$ should come from the La cations, rather than the [Mg/Ga-S]$_\infty$ anionic frameworks. But, the previous research[46] has confirmed that the trivalent La$^{3+}$ cations cannot exhibit luminescent properties (we also measured the PL spectrum of La$_2$S$_3$ (Supplementary Fig. 8), which show that La$_2$S$_3$ with the trivalent La$^{3+}$ cations have no PL property). So, these results also further indicate the nature of the divalent La$^{2+}$ (5$d^1$) cations in LaMg$_6$Ga$_6$S$_{16}$. Referencing $Li's, et al.$ first-principles calculations on the octahedral crystal-field splitting gap between the upper-lying $e_g$ and lower-lying $t_{2g}$ for the La$^{2+}$ 5$d$ orbitals in monoxide LaO, we can conclude that the green emission position at 500 nm in LaMg$_6$Ga$_6$S$_{16}$ should originate from the $d$-$d$ transition of the La$^{2+}$ within the low-coordinated octahedral crystal field, because the octahedral crystal-field splitting gap for the La$^{2+}$ 5$d$ orbitals is approximately 2.50 eV (Fig. 4e)[1,10,12,47], which is precisely consistent with the green emission position at 500 nm in LaMg$_6$Ga$_6$S$_{16}$.

Meanwhile, the decay curve of LaMg$_6$Ga$_6$S$_{16}$ under excitation at 360 nm, monitored at the peak of 500 nm at room temperature is presented in Fig. 4f. The decay curve can be fitted using a double exponential decay formula (1)[48]

$$I(t) = I_0 + A_1 exp(-t/\tau_1) + A_2 exp(-t/\tau_2) \qquad (1)$$

$$T_{ave} = (A_1\tau_1^2 + A_2\tau_2^2)/(A_1\tau_1 + A_2\tau_2) \qquad (2)$$

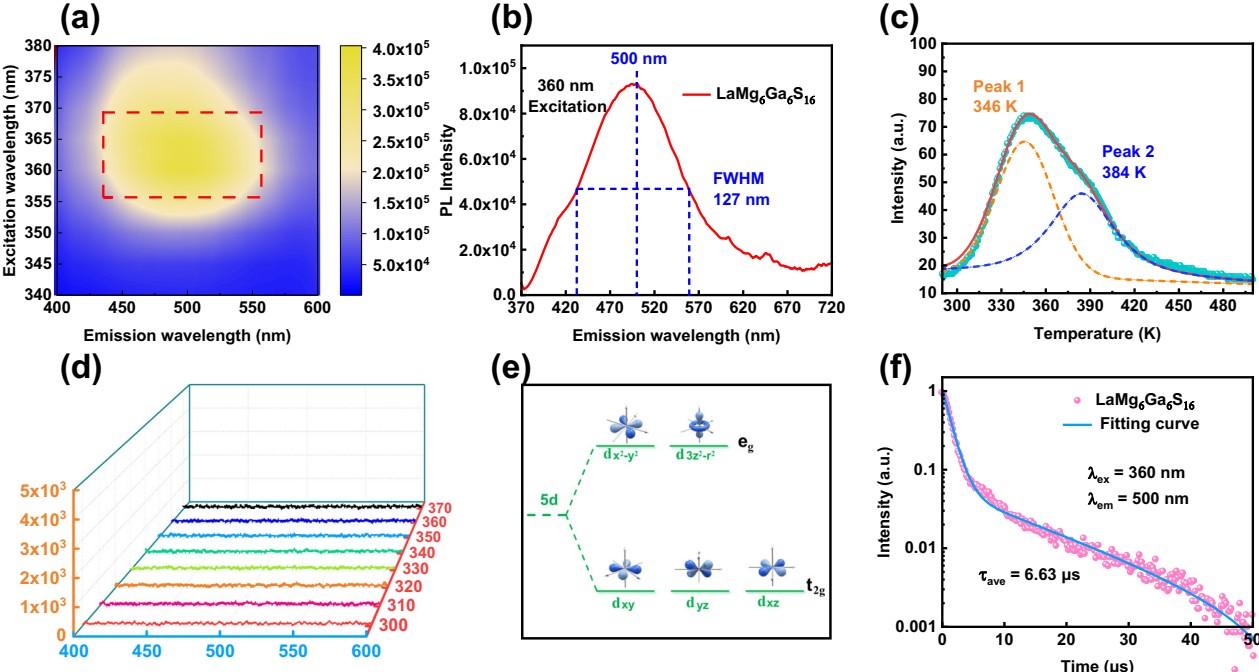

**Fig. 4 | Luminescence properties of LaMg₆Ga₆S₁₆ and CaMg₆Ga₆S₁₆.** Excitation-dependent PL spectra of LaMg₆Ga₆S₁₆ at room temperature **a**, the black dashed square indicates the ultrabroad emission range; PL emission spectra of LaMg₆Ga₆S₁₆ under 360 nm excitation at the room temperature **b**; Fitted TL spectrum of LaMg₆Ga₆S₁₆, two Gaussian bands peaking at 346 K (yellow dashed line) and 384 K (violet dashed line) **c**; Excitation-dependent PL spectra of CaMg₆Ga₆S₁₆ at room temperature **d**; Schematic diagram of the *5d* orbitals split of La²⁺ driven by the octahedral crystal field in LaMg₆Ga₆S₁₆ **e**; Room-temperature PL decay curves monitored at 500 nm and excited at 360 nm **f**.

where I(t) and I₀ denote the luminescence intensity, A₁ and A₂ are the corresponding fitting constants, and τ₁ and τ₂ are the decay time for an exponential component. As shown in Fig. 4f, by using the above fitting equation, the decay time for LaMg₆Ga₆S₁₆ can be fitted to $\tau_1 = 1.32$ μs and $\tau_2 = 15.01$ μs. According to the formula (2)[43], the value of average lifetimes ($\tau_{ave}$) was calculated to be 6.63 μs, which is similar to divalent lanthanide compounds with lifetimes in the microsecond time-range (0.5–10 μs)[1,3,48,49].

From the above discussion, LaMg₆Ga₆S₁₆ not only represents the inaugural instance of La(II)-based compounds to exhibit PL properties but also exhibits an ultrabroad-band green emission at 500 nm with FWHM of 127 nm owing to the *d-d* transition of the La²⁺ in the low-coordinated octahedral crystal field. In particular, the FWHM of 127 nm for LaMg₆Ga₆S₁₆ is larger than the developed rare earth-doped phosphor, such as CaY₂HfAl₄O₁₂:Ce³⁺ (FWHM: 120 nm)[50], β-SiAlON:Yb²⁺ (FWHM: 66 nm)[51], Li₂SrSiO₄:Pr³⁺ (FWHM: 50 nm)[52], β-SiAlON:Eu²⁺ (FWHM: 55 nm)[53], Ca₃SiO₄Cl₂:Eu²⁺ (FWHM: 59 nm)[54], Ba₂CaZn₂Si₆O₁₇:Eu²⁺ (FWHM: 80 nm)[55], Ba₃Si₆O₁₂N₂:Eu²⁺ (FWHM: 75 nm)[56], and Ca₁₀Na(PO₄)₇:Eu²⁺ (FWHM: 80 nm)[57]. More importantly, such an ultrabroad FWHM will be helpful its applications in 3D sensing, food analyzing, and other specific fields[1,40].

## NLO properties

Since LaMg₆Ga₆S₁₆ belongs to the non-centrosymmetric class and features the stable [Mg/Ga-S]∞ frameworks constructed by the NLO-active [GaS₄] tetrahedra and [MgS₆] octahedra, the NLO properties are also investigated. As a result, LaMg₆Ga₆S₁₆ shows a phase-matchable (PM) SHG response of 0.8×AgGaS₂@2090 nm (Fig. 5a and Supplementary Table 4)[58–60]. The birefringence of LaMg₆Ga₆S₁₆ was also measured on a plate-shaped crystal. It indicates that the birefringence of LaMg₆Ga₆S₁₆ at visible light is 0.041 (Fig. 5b and Supplementary Fig. 9)[61,62]. Meanwhile, the ultraviolet–vis–NIR diffusion spectrum shows that the band gap of LaMg₆Ga₆S₁₆ is 3.0 eV (Fig. 5c). The relatively large band gap causes LaMg₆Ga₆S₁₆ to generate a high powder

LIDT (~105 MW·cm⁻²)[63–65], which is more than five times that of AgGaS₂ (~20 MW·cm⁻²)[66]. Furthermore, Fourier transformation infrared (IR) (Fig. 5d) and Raman spectra (Fig. 5e) indicates that LaMg₆Ga₆S₁₆ has no obvious absorption in a wide IR range from 4000 to 500 cm⁻¹ (*i.e.*, 2.5 ~ 20 μm). Especially, compared with commercial AgGaS₂ and other important IR NLO crystals, LaMg₆Ga₆S₁₆ exhibits well-balanced NLO properties, including wide transmission region and band gaps, high LIDT, moderate birefringence as well as PM SHG responses (Fig. 5f and Supplementary Table 5). These suggest that LaMg₆Ga₆S₁₆ is also a promising IR NLO crystal. It is worth noting that the excellent NLO properties of LaMg₆Ga₆S₁₆ can, to some extent, be attributed to the particular contribution of La²⁺ cations. Since rare-earth La²⁺ cation can exhibit similar polarizability with the transition Ag⁺ and Zn²⁺ cations and comparable electropositivity with the alkali and alkaline-earth cations, LaMg₆Ga₆S₁₆ can combine the advantages of large SHG responses of transition-cations chalcogenides and large band gaps of alkali and alkaline-earth chalcogenides and achieve a better balance between large SHG response and wide band gap.

## Theoretical analysis

To better understand the structure–performance relationship, the electronic structures of LaMg₆Ga₆S₁₆ were calculated by the first-principles calculations. The calculated electronic band structure shows that LaMg₆Ga₆S₁₆ is an indirect bandgap compound with a band gap of 2.2 eV (Fig. 6a), which is smaller than the experimental value (3.0 eV) due to the limitation of using a generalized gradient approximation as the exchange- correlation functional[67]. Further, the partial densities of states of LaMg₆Ga₆S₁₆ were analyzed (Fig. 6b). It can be found that the tops of valence bands (VBs) are composed of S *3p*, Mg *2p*, and La *5d* orbitals, and the La *5d* orbitals possess the vital contribution to the top of VBs. The bottom of the conduction bonds (CBs) region is mainly Ga *4s*, Ga *4p*, Mg *3s*, Mg *3p*, La *6s*, La *5d*, and S *3p* orbitals. These results indicate that the *5d* electronic states of the La atom have a crucial effect on the band gap of the optical properties of LaMg₆Ga₆S₁₆. In

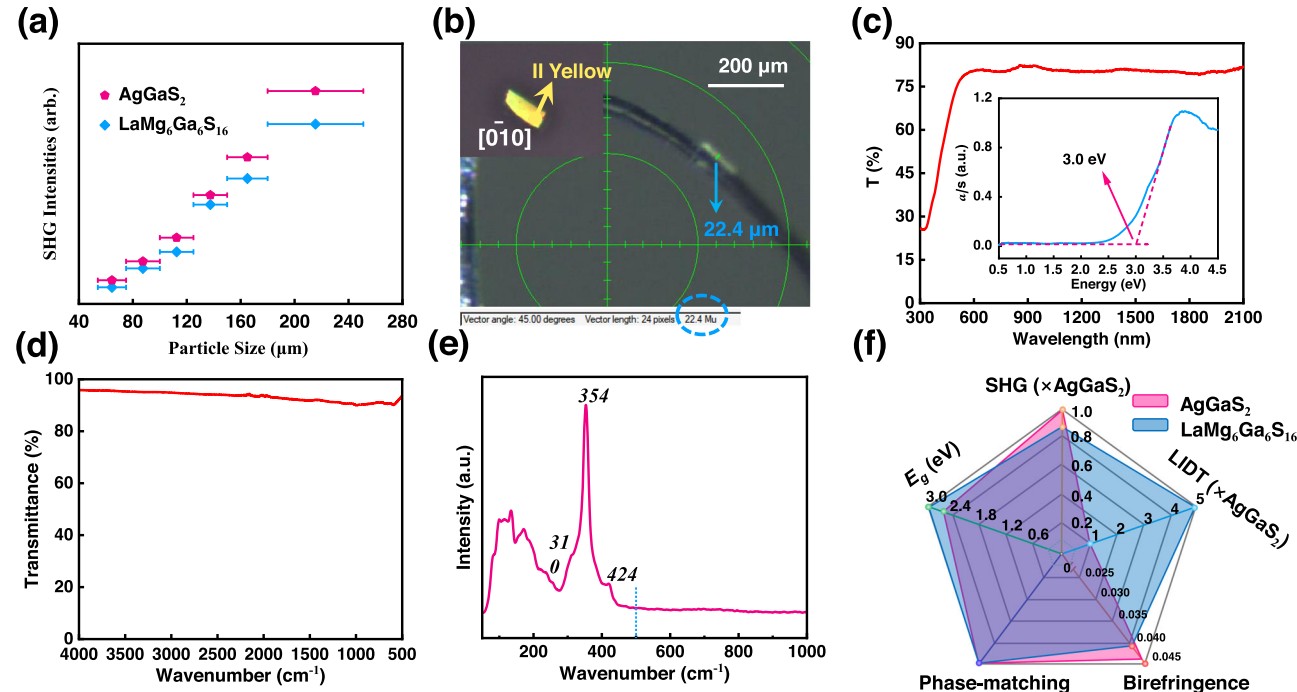

**Fig. 5 | Optical properties of LaMg₆Ga₆S₁₆.** Particle size dependence of SHG intensities of $LaMg_6Ga_6S_{16}$ (blue line) and $AgGaS_2$ (pink line) **a**, the error bars from left to right correspond to sieved crystal particle size ranges: 54–75, 75–100, 100–125, 125–150, 150–180 and 180–250 μm; Thickness of $LaMg_6Ga_6S_{16}$ crystal, inset: crystal for birefringence determination and its interference color observed in the cross-polarized light **b**; UV–vis–NIR diffuse reflectance spectrum (inset: band gap of $LaMg_6Ga_6S_{16}$ is 3.0 eV) **c**, FTIR spectrum between 4000 and 500 $cm^{-1}$ **d**, and Raman spectrum between 1000 and 50 $cm^{-1}$ **e** of $LaMg_6Ga_6S_{16}$; Well-balanced nonlinear optical properties of $LaMg_6Ga_6S_{16}$ compared to $AgGaS_2$ **f**.

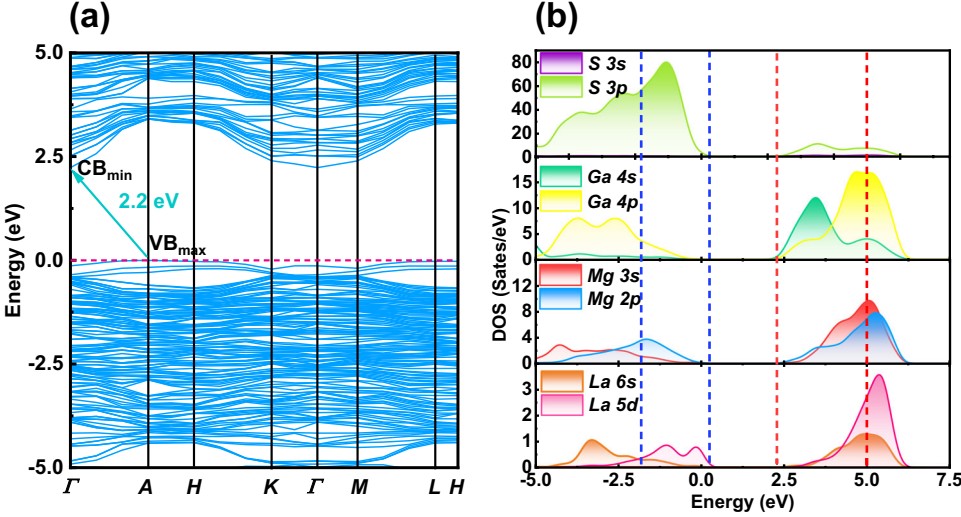

**Fig. 6 | Theoretical calculation results of LaMg₆Ga₆S₁₆.** Calculated band structure **a** and the projected density of states with the energy region from −5.0 eV to 7.5 eV **b** of $LaMg_6Ga_6S_{16}$.

addition, we also calculated SHG coefficients based on the electronic structure by the first-principles calculations. Clearly, the calculated SHG coefficients of $LaMg_6Ga_6S_{16}$ ($d_{11} = 12.27$ pm/V and $d_{22} = 4.00$ pm/V) are greater than that of the $AeMg_6Ga_6S_{16}$ (Ae = Ca, Sr, Ba) (Supplementary Table 6), suggesting divalent La make the partial contribution to SHG response of $LaMg_6Ga_6S_{16}$.

In summary, the chemically stable crystalline La(II)-chalcogenide, $LaMg_6Ga_6S_{16}$ has been synthesized by constructing the strong $[LaS_6]$ octahedra crystal field in the $[Mg/Ga-S]_\infty$ framework structure. XPS, XANES and EPR unequivocally identified the nature of the $La^{2+}$ in $LaMg_6Ga_6S_{16}$. Meanwhile, DSC, in-situ variable-temperature powder XRD and a series of solid-state reactions further illustrate its stability.

Benefiting from the unique electronic configurations of $La^{2+}$, an ultrabroad-band green emission at 500 nm with FWHM of 127 nm was discovered in $LaMg_6Ga_6S_{16}$. In particular, compared with a few synthesized divalent lanthanides organic complexes, the thermal stable divalent lanthanides inorganic compounds are still rarely researched. $LaMg_6Ga_6S_{16}$ may be able to provide some insights for the efficient syntheses of other low oxidation state lanthanide compounds.

## Methods
### Materials
La (99.9%) was purchased from Aladdin Co. Ltd. (China), MgS (99.99%), $Ga_2S_3$ (99.9%) and S (99.9%) were purchased from Beijing Hawk

Science and Technology Co. Ltd. (China), and all the reagents were used without further refinement.

## Syntheses

For the preparation of $LaMg_6Ga_6S_{16}$, reactants of La (0.2 mmol), $Ga_2S_3$ (0.6 mmol), MgS (1.2 mmol), S (0.3 mmol) were mixed and respectively loaded into graphite crucible and then they are sealed into the silica tube and flame-sealed under $10^{-3}$ Toor. The tubes were placed in a temperature-controlled furnace with the following heating process: firstly, heated to 773 K at a rate of 5 K/h and held this temperature for 10 h, then heated to 1273 K at a rate of 5 K/h and kept at that temperature for 100 h. Subsequently, the furnace was slowly cooled down to 573 K at a rate of 5 K/h. N, N−dimethylformamide (DMF) solvent was chosen to wash the products. Finally, many millimeter-level pale-yellow crystals of $LaMg_6Ga_6S_{16}$ was obtained with yields of ∼80 %, and all of them are stable under air and moisture conditions for at least 3 months. In addition, their thermal behaviors were studied by a series of solid-state reactions with the following process: their pure polycrystalline samples were firstly loaded into graphite crucibles. Then the graphite crucibles were put into silica tubes and flame-sealed under $10^{-3}$ Toor. These tubes were heated to 373 K in 10 h and kept at this temperature for about 24 h. Subsequently, they were cooled to room temperature and the mixture in the tube were thoroughly grinded and sealed into silica tubes again. The silica tubes were further heated to a higher temperature, 473 K in 10 h and kept the temperatures for 24 h. Repeating the above process with a 100 K higher calcined temperature than the last reaction.

## Structural refinement and crystal data

PXRD patterns were collected setting from the $2\theta$ range 10−70° with a step width size of 0.01° and a step time of 2 s on an automated SmartLab 3KW powder X-ray diffractometer using Cu-$K_\alpha$ radiation ($\lambda = 1.54057$ Å) radiation. The purity of compound $LaMg_6Ga_6S_{16}$ was verified by PXRD with the results as shown in Supplementary Fig. 1. To study their thermal behaviors, in-situ variable-temperature powder XRD data of $LaMg_6Ga_6S_{16}$ was collected using an SmartLab 9KW X-ray diffractometer (Supplementary Fig. 4a), meanwhile, a series of solid-state reactions with different reaction temperatures (room temperature-1323 K) were also conducted and shown in Supplementary Fig. 4b. The crystal structure of $LaMg_6Ga_6S_{16}$ was determined by single-crystal XRD on a Bruker SMART APEX III CCD diffractometer using Mo $K_\alpha$ radiation ($\lambda = 0.71073$ Å) at 297(2) K and the data was integrated with the SAINT program. All calculations were implemented with programs from the SHELXTL crystallographic software package[68]. Their crystal structures were solved by direct methods using SHELXS and refined with full-matrix least-squares methods on $F^2$ with anisotropic thermal parameters for all atoms[69]. Crystallographic data for the structure reported in this paper has been deposited with the Cambridge Crystallographic Data Centre (CCDC), under deposition number 2280420. These data can be obtained free of charge via www.ccdc.cam.ac.uk/data_request/cif, or by emailing data_request@ccdc.cam.ac.uk, or by contacting The Cambridge Crystallographic Data Centre, 12 Union Road, Cambridge CB2 1EZ, UK. Crystal data and structure refinement parameters were given in Supplementary Table 1. Some structural parameters including interatomic distances and angles, final refined atomic positions and isotropic thermal parameters are listed in Supplementary Table 2 and Supplementary Table 3, respectively.

## X-Ray photoelectron spectroscopy

The XPS (ESCALLAB250Xi, Thermo Scientific) using a monochromatized Al $K_\alpha$ source equipped with Ar ion sputtering was used for depth profiling measurements of ionic valence and composition, where the peak positions were calibrated using the C 1$s$ peak position (284.8 eV).

## Energy-dispersive spectroscopy

Microprobe elemental analyses and the elemental distribution maps were measured on a field-emission scanning electron microscope (Quanta FEG 250) made by FEI.

## Synchrotron X-ray absorption spectroscopy

The XAS measurements were carried out at the XAS Beamline at the Australian Synchrotron in Melbourne, Australia using a set of liquid nitrogen cooled Si (111) monochromator crystals. The electron beam energy is 3.0 GeV. With the associated beamline optics (Si-coated collimating mirror and Rh-coated focusing mirror), the harmonic content of the incident X-ray beam was negligible. Data was collected by using transmission mode, and the energy was calibrated using a Co foil. The beam size was about $1 \times 1$ mm. Note that a single XAS scan took about 1 h.

## Electron paramagnetic resonance spectroscopy

The EPR measurement was conducted on Bruker EMXplus-6/1 EPR spectrometer with a 9.2 GHz magnetic field.

## Thermal Analysis

The thermal behavior of $LaMg_6Ga_6S_{16}$ was performed using an HCT-4 analyzer (Beijing Henven Experimental). The sample of ∼10 mg was sealed in the customized vacuum-sealed tiny silica tubes and heated from 50 to 1300 °C at a rate of 10 °C/min. The measurements were carried out in an atmosphere of flowing $N_2$.

## Photoluminescence spectroscopy

The PL spectra were measured in room temperature using a fluorescence spectrometer (FLS-980, Edinburgh, UK). A 450 W xenon arc lamp was employed as a continuous excitation light source. The FLS980 spectrometer was configured with Red PMT photomultiplier with spectral coverage from 370 nm to 650 nm.

## Thermoluminescence spectroscopy

The TL curve was collected by the TOSL-3DS measuring instrument (PMT detector) with a heating rate of 5 °C/s after pre-irradiation for 5 min.

## Birefringence

The birefringence of $LaMg_6Ga_6S_{16}$ was measured based on a cross-polarizing microscope method with plate-shaped crystals[61]. The thickness of the used crystal is 22.4 µm for $LaMg_6Ga_6S_{16}$ (Fig. 5b), and the observed interference color is second-order yellow along [0_10] plane of the crystal in the cross-polarizing microscope (Supplementary Fig. 8). Based on the Michal-Levy chart, its retardation ($R$ value) is about 920 nm. According to the equation $R = \Delta n \times d$ (where $R$, $\Delta n$, and $d$ represent retardation, birefringence, and thickness, respectively)[60,62], the birefringence of $LaMg_6Ga_6S_{16}$ can be calculated.

## UV−vis−NIR diffuse reflectance

The UV−vis−NIR optical diffuse reflectance spectrum of $LaMg_6Ga_6S_{16}$ in the range of 300−2100 nm was measured on Shimadzu SolidSpec-3700DUV with $BaSO_4$ as a reference. The band gap was estimated on basis of the absorption spectra that was derived from the reflection spectrum using the Kubelka-Munk formula[70].

## IR and Raman spectroscopy

The IR spectrum in the range of 4000−500 $cm^{-1}$ was recorded on a Fourier transform IR spectrometer using Nicolet iS50 FT with ATR. The Raman spectrum of $LaMg_6Ga_6S_{16}$ in the range of 1000−50 $cm^{-1}$ was recorded on WITec alpha300R. The characteristic vibrations in the Raman spectrum at 424, 354, and 310 $cm^{-1}$ correspond to asymmetric and symmetric stretching vibrations of S-Ga-S and S-Mg-S modes, and peaks below 200 $cm^{-1}$ are due to the La-S and Mg-S vibrations. These

coincide with those of other related chalcogenides, such as $LaCaGa_3S_6O$ and $AeMg_6Ga_6S_{16}$ (Ae = Ca, Sr, Ba)[27,71].

## Second harmonic generation measurement

The SHG signals of $LaMg_6Ga_6S_{16}$ and benchmark $AgGaS_2$ were investigated under incident laser radiation of 2090 nm by modified Kurtz-Perry method, respectively[72]. Samples $LaMg_6Ga_6S_{16}$ and $AgGaS_2$ were sieved into several distinct particle size ranges (54–75, 75–100, 100–125, 125–150, 150–180 and 180–250 μm) for the PM measurements. The SHG signals were detected by a charge-coupled device. The second harmonic efficiency of the $LaMg_6Ga_6S_{16}$ powder was compared to that of $AgGaS_2$ powder with the same particle size.

## Laser-induced damage threshold measurement

The LIDTs of the $LaMg_6Ga_6S_{16}$ and $AgGaS_2$ powder at the particle size range of 100 – 125 μm were evaluated under using a high-power laser irradiation of 1064 nm (pulse width $\tau_p = 10$ ns) by the single-pulse method[73,74]. The measurement processes were performed by gradually increasing the laser power until the damaged spot was observed under a microscope. The damage thresholds were derived from the equation $I_{(threshold)} = E/(\pi r^2 \tau_p)$, where $E$ is the laser energy of a single pulse, $r$ is the spot radius, and $\tau_p$ is the pulse width.

## Computational methods

The electronic band structures, the partial density of states and optical properties for $LaMg_6Ga_6S_{16}$ were carried out using the CASTEP package based on density functional theory (DFT)[75]. Generalized gradient approximation (GGA) parametrized by Perdew–Burke–Ernzerhof (PBE) functional was chosen for the exchange-correlation energy, and the pseudopotential was set as norm-conserving pseudopotential (NCP)[76]. The valence electrons were set as: La $6s^25d^1$, Mg $2s^22p^63s^2$, Ga $3d^{10}4s^24p^1$, S $3s^23p^4$ for $LaMg_6Ga_6S_{16}$. The plane-wave energy cutoff value was set at 800.0 eV. The numerical integration of the Brillouin zone was performed using $2 \times 2 \times 4$ Monkhorst-Pack $\kappa$-point meshes[77]. The local-density approximation (LDA) + U approach (where U is the Hubbard energy) was adopted to deal with the strong on-site Coulomb repulsion amongst the localized La $5d$ electrons[78–80].

The SHG coefficients were calculated from the band wave functions using the so-called length-gauge formalism derived by Aversa and Sipe at a zero-frequency limit. The static second-order nonlinear susceptibilities $\chi_{\alpha\beta\gamma}^{(2)}$ can be reduced as[81–83]:

$$\chi_{\alpha\beta\gamma}^{(2)} = \chi_{\alpha\beta\gamma}^{(2)}(VE) + \chi_{\alpha\beta\gamma}^{(2)}(VH), \qquad (3)$$

Virtual-Hole (VH), Virtual-Electron (VE) and Two-Band (TB) processes play an important role in the total SHG coefficient $\chi^{(2)}$. The TB process can be neglected owing to little contribution for SHG. The formulas for calculating $\chi_{\alpha\beta\gamma}^{(2)}$ (VE) and $\chi_{\alpha\beta\gamma}^{(2)}$ (VH) are as follows:

$$\chi_{\alpha\beta\gamma}^{(2)}(VE) = \frac{e^3}{2\hbar^2 m^3} \sum_{vcc'} \int \frac{d^3k}{4\pi^3} p(\alpha\beta\gamma) \text{Im}[P_{vc}^\alpha P_{cc'}^\beta P_{c'v}^\gamma] \left( \frac{1}{\omega_{cv}^3 \omega_{vc'}^2} + \frac{2}{\omega_{vc}^4 \omega_{c'v}} \right), \quad (4)$$

$$\chi_{\alpha\beta\gamma}^{(2)}(VH) = \frac{e^3}{2\hbar^2 m^3} \sum_{vv'c} \int \frac{d^3k}{4\pi^3} p(\alpha\beta\gamma) \text{Im}[P_{vv'}^\alpha P_{v'c}^\beta P_{cv}^\gamma] \left( \frac{1}{\omega_{cv}^3 \omega_{v'c}^2} + \frac{2}{\omega_{vc}^4 \omega_{cv'}} \right), \quad (5)$$

Here, $\alpha$, $\beta$, $\gamma$ are Cartesian components, $v$ and $v'$ denote valence bands, $c$ and $c'$ refer to conduction bands, and $P(\alpha\beta\gamma)$ denotes the full permutation. The band energy difference and momentum matrix elements are denoted as $\hbar\omega_{ij}$ and $P_{ij}^\alpha$, respectively. As we know, the virtual electron (VE) progresses of occupied and unoccupied states are the main contribution to the overall SHG effect[84].

## Data availability

The representative data and extended datasets that support the findings of this study are available within the paper and its Supplementary Information files. Additional data are available from the corresponding author. The source data for Figs. 1, 3a–c, 4a–d, f, 5a, c–f, 6a, b and Supplementary Figs. 1, 3, 4a, 4b, 6, 8 are provided as a Source Data file. The X-ray crystallographic coordinates for structure reported in this study have been deposited at the Cambridge Crystall graphic Data Center (CCDC), under deposition number 2280420. These data can be obtained free of charge from The Cambridge Crystallographic Data Center via www.ccdc.cam.ac.uk/data_request/cif. Source data are provided with this paper.

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

## Acknowledgements

This work is supported by the National Natural Science Foundation of China (Grant Nos. 52322202 (H. Y.), 52172006 (H. W.), 22071179 (H. Y.), 51972230 (H. W.), 51890864 (Y. W.), 51890865 (Z. H.), Natural Science Foundation of Tianjin (Grant Nos. 20JCJQJC00060 (H. Y.) and 21JCJQJC00090 (H. W.), Tianjin University of Technology Research Innovation Project for Postgraduate Students (YJ2234 (Y. Z.)).

## Author contributions

Y.Z. performed the experiments, data analysis, and paper writing. J.C. and K.L. performed the experiments. H.W. designed and supervised the experiments. H.Y. provided major revisions of the manuscript. Z.H. supervised the optical experiments. J.W. and Y.W. helped the analyses of the crystallization process and the data. All the authors discussed the results and commented on the manuscript.

## Competing interests

The authors declare no competing interest.
