## [Peer Review File · Nature Communications]

LaMg₆Ga₆S₁₆: A Chemical Stable Divalent Lanthanide ChalcogenideREVIEWER COMMENTS

Reviewer #1 (Remarks to the Author):

This manuscript focused on the synthesis of chemically stable crystalline divalent lanthanide chalcogenide. By using octahedra coupling tetrahedra single/double chains to construct a novel octahedral crystal field, the authors claimed that they synthesized the first stable crystalline La(II)-chalcogenide, LaMg₆Ga₆S₁₆. The nature of the divalent La²⁺ cations can be identified by X-ray photoelectron spectroscopy. The authors also found that LaMg₆Ga₆S₁₆ displays a broad-band green emission at 500 nm under the excitation of 360 nm. I cannot suggest the publication of this manuscript, and some series comments are listed below, and not limited to,

1. It is known that there are complex behaviors for the chemical compositions and valence for the chalcogenide, and it commonly exist the polysulfides from condensation. It is not reasonable to give the chemical formula of LaMg₆Ga₆S₁₆ and the chemical valence of La(II). There are many different cases when one can give the chemical formula here.
2. When the authors plan to determine the chemical valence of the elements, XPS should be a very weak tool. At least the EXAFS of L edge of La should be given. Normally, La(II) should be unstable even if it can exist in this case. The authors lack enough analysis and evidence on this point.
3. The authors discussed the NLO and luminescence properties of the LaMg₆Ga₆S₁₆. Clearly, both of the performances are poor. The authors declared that this is the first time for La(II)-based compounds to exhibit luminescent properties. It is meaningless since it lacks enough analysis on the luminescence origin and the performance (broad-band green emission at 500 nm with FWHM of 127 nm) is very common and poor compared to the previous reported rare earth luminescent materials.

Reviewer #2 (Remarks to the Author):

The synthesis of divalent rare earth chalcogenids is challenging. In this manuscript, the authors reported the synthesis of a stable crystalline La(II)-chalcogenide, LaMg₆Ga₆S₁₆ in a stable three-dimensional framework constructed by octahedra and tetrahedra units. Because of the particular electronic characteristics of La²⁺ (5d¹), the compound exhibits a broad-band green emission at 500 nm. The results are interesting and meaningful to guide the fabrication of divalent rare earth chalcogenids in this family. The manuscript can be published in Nature Communications after a revision.

- (1) The compound shows a broad-band green emission at 500 nm under the excitation of 360 nm, the origin of optical property should be discussed, which could be related to the La (2+).
- (2) Except the emission band, the emission intensity should also be provided in the manuscript, which is another critical parameter for the luminescent materials, and a property comparison is encouraged.
- (3) Is there a contribution from the La (2+) to the NLO response, compared to the CaMg₆Ga₆S₁₆, SrMg₆Ga₆S₁₆, BaMg₆Ga₆S₁₆. And recent references related to wide band gap IR NLO materials, like, Adv. Opt. Mater. 2023. doi: 10.1002/adom.202300736; Mater. Horiz., 2023, 10, 619 - 624 should be cited.
- (4) The authors claimed that LaMg₆Ga₆S₁₆ is a thermally stable compound, the thermal properties should be investigated.

Reviewer #3 (Remarks to the Author):

Low-valence rare-earth compounds can exhibit many interesting electronic configurations and physicochemical properties. But their syntheses are still challengeable. In this paper, Zhang, et.al successfully synthesized a new divalent rare-earth lanthanide chalcogenide, LaMg₆Ga₆S₁₆ through constructing a stable [Mg/Ga-S]_∞ framework. The nature of the divalent La²⁺ cations and the stability of the compound are confirmed by the XPS and the in-situ variable-temperature powder XRD, respectively. The optical and NLO properties of LaMg₆Ga₆S₁₆ were also well-studied by PL and powder SHG, etc. Especially for PL measurements, it is really interesting to observe the luminescent properties for lanthanide-containing compounds. In addition, LaMg₆Ga₆S₁₆ can also

exhibit the excellent NLO properties, including moderate SHG response ($0.8 \times \text{AGS}$), wide band gap (3.0 eV) and high laser-induced damage threshold ($5 \times \text{AGS}$). It is potential as IR NLO crystal. So, I think this paper is very interesting and informative. I will recommend its acceptance after the following issues are addressed.

1) No checkcif can be found in the submitting files. The authors should upload this file to make sure their structure was refined very well.

2) For the divalent La^{2+} cation, it will contain the d electronic characteristic. Does the d electron have an effect on the band-gap of the compound? In order to better show this, the first-principle calculations for the band-gap is suggested.

3) $\text{LaMg}_6\text{Ga}_6\text{S}_{16}$ can exhibit the good NLO properties. This should also be mentioned in the abstract.

4) It is unclear that the IR spectra were recorded using spectrometer with ATR. To avoid misunderstanding, the authors should make a more detailed description on the measurement of IR spectra in experimental section.

5) Some minor grammatical errors can also be found in the manuscript. They should be corrected before publication.

Reviewer #4 (Remarks to the Author):

This manuscript reports on the first synthesis of a La-chalcogenide, $\text{LaMg}_6\text{Ga}_6\text{S}_{16}$, where La is divalent cation. As the authors noted, the divalent La chalcogenide is rare because trivalent La is very stable, and it is noteworthy that this compound is chemically stable at high temperatures. In addition, this compound showed photoluminescence and second harmonic generation.

1. From the viewpoint of solid state chemistry, the chemical stability of this divalent La chalcogenide would be very interesting. In addition, this compound can be synthesized by using conventional solid phase synthesis.

2. The optical properties are good but not excellent. In addition, the role of divalent La in the optical properties is unknown.

3. Molar percentage of La in this compound is very small, only 3%. Such small percentage would allow the anomalous valence of La, but the influence of divalent La on various properties of this compound might be very small. For example, in case of divalent Eu chalcogenides, EuO , EuS , etc, molar percentage of Eu is 50%, and the sufficiently dense Eu ion governs their eminent magnetism in these compounds.

To conclude, this study is very important in the fields of solid state chemistry, but will lack general readership, thus is not suitable for publication in this journal.

Reviewer #1:

This manuscript focused on the synthesis of chemically stable crystalline divalent lanthanide chalcogenide. By using octahedra coupling tetrahedra single/double chains to construct a novel octahedral crystal field, the authors claimed that they synthesized the first stable crystalline La(II)-chalcogenide, LaMg₆Ga₆S₁₆. The nature of the divalent La²⁺ cations can be identified by X-ray photoelectron spectroscopy. The authors also found that LaMg₆Ga₆S₁₆ displays a broad-band green emission at 500 nm under the excitation of 360 nm. I cannot suggest the publication of this manuscript, and some series comments are listed below, and not limited to,

1. It is known that there are complex behaviors for the chemical compositions and valence for the chalcogenide, and it commonly exist the polysulfides from condensation. It is not reasonable to give the chemical formula of LaMg₆Ga₆S₁₆ and the chemical valence of La(II). There are many different cases when one can give the chemical formula here.

Response: Thanks for the reviewer's comments! It is true that some reported chalcogenides contain polysulfide anions (S_n)²⁻ in their structure, such as CsGaS₃, NaNbS₆, Ba₂Ag₈S₇, Ba₂SnS₅, Cs₂Sb₂S₈, KCuCe₂S₆, NaBa₂Cu₃S₅, [Ba₄(S₂)] [ZnGa₄S₁₀]. But for these chalcogenides containing polysulfide anions, they all exhibit the S-S distances below 2.400 Å owing to the strong p_π-p_π interaction within polysulfide anions. Remarkably, all S-S distances in LaMg₆Ga₆S₁₆ exceed 3.393 Å, suggesting that the polysulfide anions do not exist in LaMg₆Ga₆S₁₆. Furthermore, bond valence calculations show that the bond valence sums (BVSs) for S are 1.82-2.18 (Table S2), which are also in agreement with its ideal oxidation states, -2. In addition, the energy-dispersive spectroscopy measurement showed the average atomic ratios of La/Mg/Ga/S were approximately equal to the theoretical ones, 3.45%, 20.69%, 20.69%, and 55.12%. All of these confirm the rationality of chemical formula of LaMg₆Ga₆S₁₆ and chemical valence of La(II).

Furthermore, in order to better characterize the chemical valence of La in LaMg₆Ga₆S₁₆, we also performed the synchrotron X-ray absorption spectroscopy (XAS) and electron paramagnetic resonance (EPR). Just as shown in Figure 3b in revised manuscript, the

La *L*-edge absorption edge of LaMg₆Ga₆S₁₆ is energy lower than that of La₂S₃, indicating a lower valence state of La in LaMg₆Ga₆S₁₆ than that in La₂S₃ (+3). This is in good agreement with the XPS results. In addition, the 5*d*¹ electronic configuration was found by EPR spectroscopy for La in LaMg₆Ga₆S₁₆, which proves the chemical valence of La(II) in LaMg₆Ga₆S₁₆. Please see more details in our response for your Question #2.

2. When the authors plan to determine the chemical valence of the elements, XPS should be a very weak tool. At least the EXAFS of L edge of La should be given. Normally, La(II) should be unstable even if it can exist in this case. The authors lack enough analysis and evidence on this point.

Response: Thanks for reviewer's comments! According to the reviewer's suggestions, we have performed the La *L*-edge X-ray absorption near-edge structure (XANES) spectrum measurements, and the results have been added in the paper. As indicated in the *L*-edge XANES spectrum (Figure. 3b), LaMg₆Ga₆S₁₆ exhibits an absorption edge with energy lower than that of La₂S₃, suggesting a lower valence state La in LaMg₆Ga₆S₁₆ than La₂S₃ (+3). This is in good agreement with the XPS results. Further, we also used electron paramagnetic resonance (EPR) to characterize the La²⁺ (5*d*¹) in the LaMg₆Ga₆S₁₆ sample. As shown in Figure 3c, a distinct EPR signal is observed for LaMg₆Ga₆S₁₆ at *g* = 1.980, which could be attributed to an unpaired electron interacting strongly with the nucleus of ¹³⁹La. The similar EPR signal was also reported on defective [K(18-crown-6)(OEt₂)][(C₅H₃(SiMe₃)₂-1,3)₃La] and [K([2.2.2]crypt)] [LaCp³] and it can be considered as the signature of the existence of La²⁺. All of these indicate the nature of the divalent La²⁺ cations in LaMg₆Ga₆S₁₆.

As for stability, it is clear that LaMg₆Ga₆S₁₆ can be obtained by a simple solid-state reaction in sealed silica tubes. And the polycrystalline LaMg₆Ga₆S₁₆ can be placed in the open air for three months without any protection. After that, no obvious changes would be observed for its powder X-ray diffraction. In addition, *in-situ* variable-temperature powder X-ray diffraction and a series of solid-state reactions in the sealed silica tubes with the different calcinated temperatures also show LaMg₆Ga₆S₁₆ has no

phase transition and decomposition when its polycrystalline sample was heated from 10 K to 1273 K (Figure S4), which also manifest that LaMg₆Ga₆S₁₆ is thermally stable. All of these confirm that LaMg₆Ga₆S₁₆ is stable in the air.

Accordingly, **the following paragraphs on the XANES and EPR measurements** have been also added at Line 7 in the left column of page 5 in the paper:

‘To better characterize the chemical valence of La in LaMg₆Ga₆S₁₆, the synchrotron X-ray absorption spectroscopy (XAS) measurements of LaMg₆Ga₆S₁₆ and La₂S₃ were performed. As indicated by the La *L*-edge X-ray absorption near-edge structure (XANES) spectra (Figure. 3b), LaMg₆Ga₆S₁₆ exhibits an absorption edge with energy lower than that of La₂S₃, indicating a lower valence state La in LaMg₆Ga₆S₁₆ than that of La₂S₃ (+3).^{32, 33} This is in good agreement with the XPS results. Further, we also used electron paramagnetic resonance (EPR) to characterize the La²⁺ (*5d*¹) in the LaMg₆Ga₆S₁₆ sample. As shown in Figure 3c, a distinct EPR signal is observed for LaMg₆Ga₆S₁₆ at *g* = 1.980, which could be attributed to an unpaired electron interacting strongly with the nucleus of ¹³⁹La.²³ The similar EPR signals have been reported on defective [K(18-crown-6)(OEt₂)][(C₅H₃(SiMe₃)₂-1,3)₃La] and [K([2.2.2]crypt)] [LaCp’’₃] and can be considered as the signature of the existence of La²⁺.¹⁰ And two new Figures as Figure 3b and Figure 3c have been added in the paper.

Figure 3. La *L*-edge normalized XANES spectra (b); EPR spectrum of LaMg₆Ga₆S₁₆ (c).

In addition, the following paragraphs have also been added in the ‘**Experimental Section**’ in the supporting information to indicate the measurement equipment and condition.

‘**Synchrotron X-ray absorption spectroscopy.** X-ray absorption spectroscopy (XAS) measurements were carried out at the XAS Beamline at the Australian Synchrotron in Melbourne, Australia using a set of liquid nitrogen cooled Si (111) monochromator crystals. The electron beam energy is 3.0 GeV. With the associated beamline optics (Si-coated collimating mirror and Rh-coated focusing mirror), the harmonic content of the incident X-ray beam was negligible. Data was collected by using transmission mode, and the energy was calibrated using a Co foil. The beam size was about 1×1mm. Note that a single XAS scan took about 1 h.’

‘**Electron Paramagnetic Resonance Spectroscopy.** The electron paramagnetic resonance (EPR) measurement was conducted on Bruker EMXplus-6/1 EPR spectrometer with a 9.2 GHz magnetic field.’

The new references ‘32. Yuan X, *et al.* Hydrolase mimic via second coordination sphere engineering in metal-organic frameworks for environmental remediation. *Nat. Commun.* **14**, 5974 (2023).; 33. Cui T, *et al.* Engineering Dual Single-Atom Sites on 2D Ultrathin N-doped Carbon Nanosheets Attaining Ultra-Low-Temperature Zinc-Air Battery. *Angew. Chem. Int. Ed.* **61**, e202115219 (2022)’, were also added as References 32 and 33.

3. The authors discussed the NLO and luminescence properties of the LaMg₆Ga₆S₁₆. Clearly, both of the performances are poor. The authors declared that this is the first time for La(II)-based compounds to exhibit luminescent properties. It is meaningless since it lacks enough analysis on the luminescence origin and the performance (broad-band green emission at 500 nm with FWHM of 127 nm) is very common and poor compared to the previous reported rare earth luminescent materials.

Response: Thanks for the reviewer’s comments!

As shown in Figure 5, LaMg₆Ga₆S₁₆ has exhibited excellent NLO properties, including

i) a large SHG response ($0.8 \times \text{AgGaS}_2$), ii) wide band gap ($E_g = 3.0 \text{ eV}$), iii) broad IR transmission covering two important atmospheric windows, 3–5 μm and 8–12 μm , iv) a moderate birefringence ($\Delta n = 0.04$), v) ease for crystal growth because of the congruently melting property, and vi) stable physicochemical properties. **These properties can well satisfy the widely-considered requirements of an excellent IR NLO crystal**, including i) large SHG responses ($d_{ij} > 0.5 \text{ AgGaS}_2$); ii) wide band gap ($E_g > 3.0 \text{ eV}$); iii) broad transmission window (3–14 μm); iv) a moderate birefringence ($0.03 > \Delta n > 0.1$) to meet the phase-matching condition; and v) good thermal and environmental stability (*Angew. Chem. Int. Ed.* **59**, 7514-7520 (2020); *Adv. Funct. Mater.* **32**, 2200231 (2022); *Coord. Chem. Rev.* **470**, 214706 (2022)).

In addition, according to the reviewer's suggestion, we also performed the thermoluminescence (TL) measurement and added the analysis on the luminescence origin and the performance comparison to other reported rare earth luminescent materials. That shows *LaMg₆Ga₆S₁₆ can exhibit the wider ultrabroad emission at 500 nm with a full width at half maximum (FWHM) of 127 nm than other reported rare earth luminescent materials, such as CaY₂HfAl₄O₁₂:Ce³⁺ (FWHM: 120 nm), β -SiAlON:Yb²⁺ (FWHM: 66 nm), Li₂SrSiO₄:Pr³⁺ (FWHM: about 50 nm), β -SiAlON:Eu²⁺ (FWHM: 55 nm), Ca₃SiO₄Cl₂:Eu²⁺ (FWHM: 59 nm), Ba₂CaZn₂Si₆O₁₇:Eu²⁺ (FWHM: 80 nm), Ba₃Si₆O₁₂N₂:Eu²⁺ (FWHM: 75 nm), and Ba₂SiO₄:Eu²⁺ (FWHM: 80 nm).* Such an ultrabroad emission band covers almost the whole visible light region and is interesting for the warm-white light-emitting diodes (LED). So, LaMg₆Ga₆S₁₆ may be interesting for the ultrabroad emission luminescent materials.

In addition, a thermoluminescence (TL) measurement and the following paragraphs to illustrate the luminescence origin have been added at Line 18 in the right column of page 6 in the paper: 'Further, to investigate the origin of luminescence properties, the thermoluminescence (TL) measurement of LaMg₆Ga₆S₁₆ is performed. As shown in Figure 4c, the sample shows a broad TL glow curve in the range from 290 K to 450 K, indicating a continuous distribution of trap depths. By fitting the TL curve with two Gaussian bands peaking at 346 K and 384 K, the characteristic trap depths (E_T) were estimated to be 0.69 and 0.77 eV by using the crude relationship $E_T = T_m/500 \text{ eV}$, where

T_m represents the temperature (K) of the TL fitting peak.^{43, 44} In view of the EPR results ($g = 2.008$), it is evident that the two trap depths originate from the intrinsic defects, corresponding to the S vacancies defect.^{45, 46} Based on these studies, a possible mechanism for luminescence is proposed and displayed in Figure 4d. Initially after photoexcitation, the excitons in the excited states of host will be trapped by the intrinsic intermediate defect states (process 1). The trapped carriers/energies will recombine non-radiatively if no La^{2+} exists in the host (process 2). However, owing to the presence of La^{2+} in $\text{LaMg}_6\text{Ga}_6\text{S}_{16}$ and La^{2+} $5d$ orbitals split in the low-coordinated octahedral crystal field (Figure 4e),^{1, 12, 25, 47} the electrons captured by trap states directly reach the e_g of La^{2+} $5d$ orbitals by efficient energy transfer (process 3). While these electrons occur transition from upper-lying e_g to lower-lying t_{2g} of La^{2+} $5d$ orbitals in the low-coordinated octahedral crystal field according to Dorenbos theory, accompanied by the emission of green light at 500 nm (corresponding to 2.48 eV) (process 4). This result is comparable to the previously reported octahedral crystal-field splitting gap of approximately 2.50 eV for the La $5d$ orbitals in monoxide LaO .¹²

The paragraph on the performance comparison between $\text{LaMg}_6\text{Ga}_6\text{S}_{16}$ and other reported rare earth luminescent materials was added at Line 13 in the right column of page 7: ‘From the above discussion, $\text{LaMg}_6\text{Ga}_6\text{S}_{16}$ not only represents the first La(II)-based compounds to exhibit photoluminescence properties but also exhibits an ultrabroad-band green emission at 500 nm with FWHM of 127 nm owing to the $d-d$ transition of the La^{2+} in the low-coordinated octahedral crystal field. In particular, the FWHM of 127 nm for $\text{LaMg}_6\text{Ga}_6\text{S}_{16}$ is larger than the developed rare earth-doped phosphor, such as $\text{CaY}_2\text{HfAl}_4\text{O}_{12}:\text{Ce}^{3+}$ (FWHM: 120 nm)⁵⁰, $\beta\text{-SiAlON}:\text{Yb}^{2+}$ (FWHM: 66 nm)⁵¹, $\text{Li}_2\text{SrSiO}_4:\text{Pr}^{3+}$ (FWHM: 50 nm)⁵², $\beta\text{-SiAlON}:\text{Eu}^{2+}$ (FWHM: 55 nm)⁵³, $\text{Ca}_3\text{SiO}_4\text{Cl}_2:\text{Eu}^{2+}$ (FWHM: 59 nm)⁵⁴, $\text{Ba}_2\text{CaZn}_2\text{Si}_6\text{O}_{17}:\text{Eu}^{2+}$ (FWHM: 80 nm)⁵⁵, $\text{Ba}_3\text{Si}_6\text{O}_{12}\text{N}_2:\text{Eu}^{2+}$ (FWHM: 75 nm)⁵⁶, and $\text{Ca}_{10}\text{Na}(\text{PO}_4)_7:\text{Eu}^{2+}$ (FWHM: 80 nm)⁵⁷. More importantly, such an ultrabroad FWHM will be helpful its applications in 3D sensing, food analyzing, and other specific fields.^{1, 41} And three new Figures as Figure 4c, Figure 4d and Figure 4e have been added in the paper.

Figure 4. TL spectrum (c) and schematic illustration of the proposed luminescent processes (d) of $\text{LaMg}_6\text{Ga}_6\text{S}_{16}$; Schematic diagram of the $5d$ orbitals split of La^{2+} driven by the octahedral crystal field in $\text{LaMg}_6\text{Ga}_6\text{S}_{16}$ (e).

In addition, the following paragraph has also been added in the ‘**Experimental Section**’ in the supporting information to indicate the measurement equipment and condition. ‘**Thermoluminescence measurement.** The TL curves were collected by the TOSL-3DS measuring instrument (PMT detector) with a heating rate of $5\text{ }^\circ\text{C/s}$ after pre-irradiation for 5 min.’

Meanwhile, we have also added the references ‘41. Huang S, Shang M, Yan Y, Wang Y, Dang P, Lin J. Ultra-Broad band Green-Emitting Phosphors without Cyan Gap Based on Double-Heterovalent Substitution Strategy for Full-Spectrum WLED Lighting. *Laser Photonics Rev.* **16**, 2200473 (2022).; 50. Chan J, Devakumar B, Li W, Ma N, Huang X, Lee AF. Full-spectrum solid-state white lighting with high color rendering index exceeding 96 based on a bright broadband green-emitting phosphor. *Appl. Mater. Today* **27**, 101439 (2022).; 51. Liu L, *et al.* Photoluminescence properties of beta-SiAlON:Yb $^{2+}$, a novel green-emitting phosphor for white light-emitting diodes. *Sci. Technol. Adv. Mater.* **12**, 034404 (2011).; 52. Chen J, Guo C, Yang Z, Li T, Zhao J, McKittrick J. Li $_2$ SrSiO $_4$:Ce $^{3+}$, Pr $^{3+}$ Phosphor with Blue, Red, and Near - Infrared Emissions Used for Plant Growth LED. *J. Am. Ceram. Soc.* **99**, 218-225 (2015).; 53. Hirosaki N, Xie R, Kimoto K, Sekiguchi T, Yamamoto Y, Suehiro T, Mitomo M. Characterization and properties of green-emitting β -SiAlON:Eu $^{2+}$ powder phosphors for white light-emitting diodes. *Appl. Phys. Lett.* **86**, 211905 (2005).; 54. Liu J, Lian H, Sun J, Shi C. Characterization and Properties of Green Emitting Ca $_3$ SiO $_4$ Cl $_2$:Eu $^{2+}$

Powder Phosphor for White Light-emitting Diodes. *Chem. Lett.* **34**, 1340-1341 (2005).; 55. Annadurai G, Kennedy S, Sivakumar V. Luminescence properties of a novel green emitting $\text{Ba}_2\text{CaZn}_2\text{Si}_6\text{O}_{17}:\text{Eu}^{2+}$ phosphor for white light-Emitting diodes applications. *Superlattices Microstruct.* **93**, 57-66 (2016).; 56. Li C, Chen H, Xu S. $\text{Ba}_3\text{Si}_6\text{O}_{12}\text{N}_2:\text{Eu}^{2+}$ green-emitting phosphor for white light emitting diodes: Luminescent properties optimization and crystal structure analysis. *Optik* **126**, 499-502 (2015).; 57. Zhao J, Wu Y, Liang Y, Liu M, Yang F, Xia Z. A novel green-emitting phosphor $\text{Ca}_{10}\text{Na}(\text{PO}_4)_7:\text{Eu}^{2+}$ for near ultraviolet white light-emitting diodes. *Opt. Mater.* **35**, 1675-1678 (2013)' as References 41 and 50-57.

Reviewer #2 (Remarks to the Author):

The synthesis of divalent rare earth chalcogenids is challenging. In this manuscript, the authors reported the synthesis of a stable crystalline La(II)-chalcogenide, $\text{LaMg}_6\text{Ga}_6\text{S}_{16}$ in a stable three-dimensional framework constructed by octahedra and tetrahedra units. Because of the particular electronic characteristics of La^{2+} ($5d^1$), the compound exhibits a broad-band green emission at 500 nm. The results are interesting and meaningful to guide the fabrication of divalent rare earth chalcogenids in this family. The manuscript can be published in Nature Communications after a revision.

1. The compound shows a broad-band green emission at 500 nm under the excitation of 360 nm, the origin of optical property should be discussed, which could be related to the La ($2+$).

Response: Thanks for the reviewer's kind comments and good suggestion! According to the reviewer's suggestion, to reveal the origin of photoluminescence properties, we performed the thermoluminescence (TL) measurement of $\text{LaMg}_6\text{Ga}_6\text{S}_{16}$, and the results have been added in the paper (Figure 4c-e). Please see more details in our response for Question 3 of Review #1.

2. Except the emission band, the emission intensity should also be provided in the manuscript, which is another critical parameter for the luminescent materials, and a property comparison is encouraged.

Response: Thanks for reviewer's kind comments and good suggestions! According to the reviewer's suggestions, the emission intensity of LaMg₆Ga₆S₁₆ have added in Figure 4b, and new Figure 4b has been added in the article.

In addition, we also further compared the luminescent property of LaMg₆Ga₆S₁₆ with other reported luminescent materials. Clearly, LaMg₆Ga₆S₁₆ exhibits a wider ultrabroad-band green emission at 500 nm with FWHM of 127 nm owing to the *d-d* transition of the La²⁺. The FWHM of 127 nm for LaMg₆Ga₆S₁₆ is larger than the developed rare earth-doped phosphor, such as CaY₂HfAl₄O₁₂:Ce³⁺ (FWHM: 120 nm), β-SiAlON:Yb²⁺ (FWHM: 66 nm), Li₂SrSiO₄:Pr³⁺ (FWHM: 50 nm), β-SiAlON:Eu²⁺ (FWHM: 55 nm), Ca₃SiO₄Cl₂:Eu²⁺ (FWHM: 59 nm), Ba₂CaZn₂Si₆O₁₇:Eu²⁺ (FWHM: 80 nm), Ba₃Si₆O₁₂N₂:Eu²⁺ (FWHM: 75 nm), and Ca₁₀Na(PO₄)₇:Eu²⁺ (FWHM: 80 nm). More importantly, such an ultrabroad emission band covers almost the whole visible light region and is interesting for the human-centric full-visible-spectrum lighting, such as 3D sensing, food analyzing, and other specific fields. Meanwhile, this characteristic green emission under 360 nm excitation endows LaMg₆Ga₆S₁₆ with the potential light-emitting diode application under the excitation of commercial near ultraviolet chips. This research provides some new insights for the efficient syntheses of other low oxidation state lanthanide compounds and further promotes the development of new La(II)-based compounds in photoluminescence applications.

Accordingly, the following paragraph have been added at Line 11 in the right column of page 6 in the paper: 'The ultrabroad emission band cover almost the whole visible light region and could find applications in the field of human-centric full-visible-spectrum lighting.⁴¹ Meanwhile, this characteristic green emission under 360 nm excitation endows LaMg₆Ga₆S₁₆ with the potential light-emitting diode application under the excitation of commercial near ultraviolet chips.⁴²' and at Line 18 in the right column of page 7 in the paper: 'In particular, the FWHM of 127 nm for LaMg₆Ga₆S₁₆ is larger than the developed rare earth-doped phosphor, such as CaY₂HfAl₄O₁₂:Ce³⁺ (FWHM: 120 nm)⁵⁰, β-SiAlON:Yb²⁺ (FWHM: 66 nm)⁵¹, Li₂SrSiO₄:Pr³⁺ (FWHM: 50 nm)⁵², β-SiAlON:Eu²⁺ (FWHM: 55 nm)⁵³, Ca₃SiO₄Cl₂:Eu²⁺ (FWHM: 59 nm)⁵⁴, Ba₂CaZn₂Si₆O₁₇:Eu²⁺ (FWHM: 80 nm)⁵⁵, Ba₃Si₆O₁₂N₂:Eu²⁺ (FWHM: 75 nm)⁵⁶, and

$\text{Ca}_{10}\text{Na}(\text{PO}_4)_7:\text{Eu}^{2+}$ (FWHM: 80 nm)⁵⁷. More importantly, such an ultrabroad FWHM will be helpful its applications in 3D sensing, food analyzing, and other specific fields.^{1, 41,}

Meanwhile, we have also added the references ‘41. Huang S, Shang M, Yan Y, Wang Y, Dang P, Lin J. Ultra-Broad band Green-Emitting Phosphors without Cyan Gap Based on Double-Heterovalent Substitution Strategy for Full-Spectrum WLED Lighting. *Laser Photonics Rev.* **16**, 2200473 (2022).; 50. Chan J, Devakumar B, Li W, Ma N, Huang X, Lee AF. Full-spectrum solid-state white lighting with high color rendering index exceeding 96 based on a bright broadband green-emitting phosphor. *Appl. Mater. Today* **27**, 101439 (2022).; 51. Liu L, *et al.* Photoluminescence properties of beta-SiAlON:Yb²⁺, a novel green-emitting phosphor for white light-emitting diodes. *Sci. Technol. Adv. Mater.* **12**, 034404 (2011).; 52. Chen J, Guo C, Yang Z, Li T, Zhao J, McKittrick J. Li₂SrSiO₄:Ce³⁺, Pr³⁺ Phosphor with Blue, Red, and Near - Infrared Emissions Used for Plant Growth LED. *J. Am. Ceram. Soc.* **99**, 218-225 (2015).; 53. Hirosaki N, Xie R, Kimoto K, Sekiguchi T, Yamamoto Y, Suehiro T, Mitomo M. Characterization and properties of green-emitting β -SiAlON:Eu²⁺ powder phosphors for white light-emitting diodes. *Appl. Phys. Lett.* **86**, 211905 (2005).; 54. Liu J, Lian H, Sun J, Shi C. Characterization and Properties of Green Emitting Ca₃SiO₄Cl₂:Eu²⁺ Powder Phosphor for White Light-emitting Diodes. *Chem. Lett.* **34**, 1340-1341 (2005).; 55. Annadurai G, Kennedy S, Sivakumar V. Luminescence properties of a novel green emitting Ba₂CaZn₂Si₆O₁₇:Eu²⁺ phosphor for white light-Emitting diodes applications. *Superlattices Microstruct.* **93**, 57-66 (2016).; 56. Li C, Chen H, Xu S. Ba₃Si₆O₁₂N₂:Eu²⁺ green-emitting phosphor for white light emitting diodes: Luminescent properties optimization and crystal structure analysis. *Optik* **126**, 499-502 (2015).; 57. Zhao J, Wu Y, Liang Y, Liu M, Yang F, Xia Z. A novel green-emitting phosphor Ca₁₀Na(PO₄)₇:Eu²⁺ for near ultraviolet white light-emitting diodes. *Opt. Mater.* **35**, 1675-1678 (2013).’, as Reference 41, 50-57, and the following reference numbers have also been updated.

3. Is there a contribution from the La (2+) to the NLO response, compared to the CaMg₆Ga₆S₁₆, SrMg₆Ga₆S₁₆, BaMg₆Ga₆S₁₆. And recent references related to wide band

gap IR NLO materials, like, Adv. Opt. Mater. 2023. doi: 10.1002/adom.202300736; Mater. Horiz., 2023, 10, 619 - 624 should be cited.

Response: Thanks for the reviewer’s kind suggestions! To investigate the contribution of divalent La on the NLO properties, the electronic structures of LaMg₆Ga₆S₁₆ were calculated by the first-principles calculations. Clearly, compared with the reported AeMg₆Ga₆S₁₆ (Ae = Ca, Sr, Ba), the calculated SHG coefficients of LaMg₆Ga₆S₁₆ ($d_{11} = 12.27$ pm/V and $d_{22} = 4.00$ pm/V) are greater than that of the AeMg₆Ga₆S₁₆ (Ae = Ca, Sr, Ba) (Table S6), suggesting divalent La make the partial contribution to NLO response of LaMg₆Ga₆S₁₆. Meanwhile, near the Fermi surface in LaMg₆Ga₆S₁₆, the divalent La *5d* orbitals possess the vital contribution to the top of valence band and the bottom of the conduction bands (Figure 6). These indicate that the *5d* electronic states of divalent La have the vital contribution to NLO response and band gap of LaMg₆Ga₆S₁₆ because the optical properties of a material are mainly affected by the electronic transitions between the energy levels close to the forbidden band.

Accordingly, the following paragraphs have been added at Line 10 in the left column of page 9: ‘In addition, we also calculated SHG coefficients based on the electronic structure by the first-principles calculations. Clearly, the calculated SHG coefficients of LaMg₆Ga₆S₁₆ ($d_{11} = 12.27$ pm/V and $d_{22} = 4.00$ pm/V) are greater than that of the AeMg₆Ga₆S₁₆ (Ae = Ca, Sr, Ba), suggesting divalent La make the partial contribution to SHG response of LaMg₆Ga₆S₁₆.’, and a new Table S6 has been added in the paper.

Table S6. The space groups and SHG coefficients of LaMg₆Ga₆S₁₆ and AeMg₆Ga₆S₁₆ (Ae = Ca, Sr, Ba).

Compounds	Space groups	d_{ij} (pm/V)	
LaMg ₆ Ga ₆ S ₁₆	P -6	$d_{11} = 12.27$	$d_{22} = 4.00$
CaMg ₆ Ga ₆ S ₁₆ .	P -6	$d_{11} = 9.93$	$d_{22} = 3.82$
SrMg ₆ Ga ₆ S ₁₆ .	P -6	$d_{11} = 10.13$	$d_{22} = 3.83$
BaMg ₆ Ga ₆ S ₁₆ .	P -6	$d_{11} = 10.19$	$d_{22} = 3.87$

In addition, we have added the references ‘63. Zhou J, Wang L, Chu Y, Wang H, Pan S, Li J. Na₃SiS₃F: A Wide Bandgap Fluorothiosilicate with Unique [SiS₃F] Unit and High Laser-Induced Damage Threshold. *Adv. Opt. Mater.* **11**, 2300736 (2023).; 64. Zhou J, Fan Z, Zhang K, Yang Z, Pan S, Li J. Rb₂CdSi₄S₁₀: novel [Si₄S₁₀] T2-supertetrahedra-contained infrared nonlinear optical material with large band gap. *Mater. Horiz.* **10**, 619-624 (2023).; 65. Wang P, *et al.* The Combination of Structure Prediction and Experiment for the Exploration of Alkali-Earth Metal-Contained Chalcopyrite-Like IR Nonlinear Optical Material. *Adv. Sci.* **9**, 2106120 (2022).’, as References 63-65, and the following reference numbers have also been updated.

4. The authors claimed that LaMg₆Ga₆S₁₆ is a thermally stable compound, the thermal properties should be investigated.

Response: Thanks a lot for the reviewer’s kind comments and good suggestion! According to the reviewer’s suggestion, the thermal behavior of LaMg₆Ga₆S₁₆ was further studied by differential thermal scanning (DSC) measurements. The measurement results have been added in the paper. Clearly, only one endothermic peak at 1140 °C was observed on the heating DSC curve (Figure S3), suggesting that LaMg₆Ga₆S₁₆ did not undergo the decomposition and structural phase transitions when the temperature was increased from room temperature to 1140 °C. Moreover, *in-situ* variable-temperature powder X-ray diffraction and a series of solid-state reactions in the sealed silica tubes with the different calcinated temperatures show LaMg₆Ga₆S₁₆ has no phase transition when its polycrystalline sample was heated from 10 K to 1273 K (Figure S4), which further manifest that LaMg₆Ga₆S₁₆ is thermally stable.

Accordingly, the following sentences have been also added at Line 30 in the left column of page 5 in the paper: ‘Further, the thermal behavior of LaMg₆Ga₆S₁₆ was studied by differential thermal scanning (DSC) measurements. Clearly, only one endothermic peak at 1140 °C was observed on the heating DSC curve (Figure S3), suggesting that LaMg₆Ga₆S₁₆ did not undergo the decomposition and structural phase transitions when the temperature was increased from room temperature to 1140 °C. Moreover, *in-situ* variable-temperature powder X-ray diffraction and a series of solid-state reactions in

the sealed silica tubes with the different calcinated temperatures show $\text{LaMg}_6\text{Ga}_6\text{S}_{16}$ has no phase transition when its polycrystalline sample was heated from 10 K to 1273 K (Figure S4), which also manifest that $\text{LaMg}_6\text{Ga}_6\text{S}_{16}$ is thermally stable.’ And one new Figure as Figure S3 has been added in the supporting information.

The following paragraph has also been added in the ‘**Experimental Section**’ in the supporting information to indicate the measurement equipment and condition.

‘**Thermal Analysis.** The thermal behavior of $\text{LaMg}_6\text{Ga}_6\text{S}_{16}$ was performed using an HCT-4 analyzer (Beijing Henven Experimental). The sample of ~ 10 mg was sealed in the customized vacuum-sealed tiny silica tubes and heated from 50 to 1300 °C at a rate of 10 °C/min. The measurements were carried out in an atmosphere of flowing N_2 .’

Figure S3. The DSC curve of $\text{LaMg}_6\text{Ga}_6\text{S}_{16}$.

Reviewer #3 (Remarks to the Author):

Low-valence rare-earth compounds can exhibit many interesting electronic configurations and physicochemical properties. But their syntheses are still challengeable. In this paper, Zhang, et.al successfully synthesized a new divalent rare-earth lanthanide chalcogenide, $\text{LaMg}_6\text{Ga}_6\text{S}_{16}$ through constructing a stable $[\text{Mg}/\text{Ga}-\text{S}]_\infty$ framework. The nature of the divalent La^{2+} cations and the stability of the compound are confirmed by the XPS and *the in-situ* variable-temperature powder XRD, respectively. The optical and NLO properties of $\text{LaMg}_6\text{Ga}_6\text{S}_{16}$ were also well-studied

by PL and powder SHG, etc. Especially for PL measurements, it is really interesting to observe the luminescent properties for lanthanide-containing compounds. In addition, LaMg₆Ga₆S₁₆ can also exhibit the excellent NLO properties, including moderate SHG response (0.8 × AGS), wide band gap (3.0 eV) and high laser-induced damage threshold (5 × AGS). It is potential as IR NLO crystal. So, I think this paper is very interesting and informative. I will recommend its acceptance after the following issues are addressed.

1. No cif and checkcif can be found in the submitting files. The authors should upload these files to make sure their structure was refined very well.

Response: Thanks for reviewer's kind comments and good suggestions! According to the reviewer's suggestions, we have added the Checkcif file of LaMg₆Ga₆S₁₆ in the Supporting information.

2. For the divalent La²⁺ cation, it will contain the d electronic characteristic. Does the d electron have an effect on the band-gap of the compound? In order to better show this, the first-principle calculations for the band-gap will be suggested.

Response: Thanks for reviewer's good suggestion! According to the reviewer's suggestion, we have calculated the band gap and the electronic densities of states of LaMg₆Ga₆S₁₆ based on the first-principles calculations. The calculated result that LaMg₆Ga₆S₁₆ is indirect band gap with the calculated band gap of 2.2 eV (Figure 6a), which is smaller than the experimental value (3.0 eV) due to the discontinuity of exchange-correlation energy. Further, the partial densities of states of LaMg₆Ga₆S₁₆ were analyzed (Figure 6b). It can be found that the tops of valence bands (VBs) are composed of S 3p, Mg 2p, and La 5d orbitals, and the La 5d orbitals possess the vital contribution to the top of VBs. The bottom of the conduction bands (CBs) region is mainly Ga 4s, Ga 4p, Mg 3s, Mg 3s, La 6s, La 5p, La 5d, and S 3p orbitals. These indicate that the 5d electronic states of La atom do affect the band gap of the compound. Accordingly, the following paragraph have been added at Line 11 in the right column of page 8 in the paper: 'To better understand the structure–performance relationship, the electronic structures of LaMg₆Ga₆S₁₆ were calculated by the first-principles calculations. The calculated electronic band structure shows that LaMg₆Ga₆S₁₆ is an

indirect bandgap compound with a band gap of 2.2 eV (Figure 6a), which is smaller than the experimental value (3.0 eV) due to the limitation of using a generalized gradient approximation as the exchange-correlation functional.⁶⁷ Further, the partial densities of states of LaMg₆Ga₆S₁₆ were analyzed (Figure 6b). It can be found that the tops of valence bands (VBs) are composed of S 3*p*, Mg 2*p*, and La 5*d* orbitals, and the La 5*d* orbitals possess the vital contribution to the top of VBs. The bottom of the conduction bands (CBs) region is mainly Ga 4*s*, Ga 4*p*, Mg 3*s*, Mg 3*p*, La 6*s*, La 5*d*, and S 3*p* orbitals. These results indicate that the 5*d* electronic states of the La atom have a crucial effect on the band gap of the optical properties of LaMg₆Ga₆S₁₆. And two new Figures as Figure 6a and Figure 6b have been added in the paper.

Figure 6. Calculated band structure (a) and the projected density of states (b) of LaMg₆Ga₆S₁₆.

In addition, we have added the following paragraphs on “Computation methods” in the paper. ‘The electronic band structures, the partial density of states and optical properties for LaMg₆Ga₆S₁₆ were carried out using the CASTEP package based on density functional theory (DFT).⁷⁷ Generalized gradient approximation (GGA) parametrized by Perdew-Burke-Ernzerhof (PBE) functional was chosen for the exchange-correlation energy, and the pseudopotential was set as norm-conserving pseudopotential (NCP).⁷⁸ The valence electrons were set as: La 6*s*²5*d*¹, Mg 2*s*²2*p*⁶3*s*², Ga 3*d*¹⁰4*s*²4*p*¹, S 3*s*²3*p*⁴ for LaMg₆Ga₆S₁₆. The plane-wave energy cutoff value was set at 800.0 eV. The numerical

integration of the Brillouin zone was performed using $2 \times 2 \times 4$ Monkhorst-Pack κ -point meshes.⁷⁹ The Hubbard U model formalism formulated was adopted for the evaluation of the strong on-site Coulomb repulsion amongst the localized La 5d electrons.^{80, 81, 82}

The SHG coefficients were calculated from the band wave functions using the so-called length-gauge formalism derived by Aversa and Sipe at a zero-frequency limit. The static second-order nonlinear susceptibilities $\chi_{\alpha\beta\gamma}^{(2)}$ can be reduced as:^{83, 84, 85}

$$\chi_{\alpha\beta\gamma}^{(2)} = \chi_{\alpha\beta\gamma}^{(2)}(\text{VE}) + \chi_{\alpha\beta\gamma}^{(2)}(\text{VH}) \quad (1),$$

Virtual-Hole (VH), Virtual-Electron (VE) and Two-Band (TB) processes play an important role in the total SHG coefficient $\chi^{(2)}$. The TB process can be neglected owing to little contribution for SHG. The formulas for calculating $\chi_{\alpha\beta\gamma}^{(2)}(\text{VE})$ and $\chi_{\alpha\beta\gamma}^{(2)}(\text{VH})$ are as follows:

$$\chi_{\alpha\beta\gamma}^{(2)}(\text{VE}) = \frac{e^3}{2\hbar^2 m^3} \sum_{vcc'} \int \frac{d^3k}{4\pi^3} p(\alpha\beta\gamma) \text{Im}[P_{vc}^\alpha P_{cc'}^\beta P_{cv}^\gamma] \left(\frac{1}{\omega_{cv}^3 \omega_{v'c}^2} + \frac{2}{\omega_{v'c}^4 \omega_{cv}} \right) \quad (2),$$

$$\chi_{\alpha\beta\gamma}^{(2)}(\text{VH}) = \frac{e^3}{2\hbar^2 m^3} \sum_{vv'c} \int \frac{d^3k}{4\pi^3} p(\alpha\beta\gamma) \text{Im}[P_{vv'}^\alpha P_{v'c}^\beta P_{cv}^\gamma] \left(\frac{1}{\omega_{cv}^3 \omega_{v'c}^2} + \frac{2}{\omega_{v'c}^4 \omega_{cv}} \right) \quad (3),$$

Here, α, β, γ are Cartesian components, v and v' denote valence bands, c and c' refer to conduction bands, and $P(\alpha\beta\gamma)$ denotes the full permutation. The band energy difference and momentum matrix elements are denoted as $\hbar\omega_{ij}$ and P_{ij}^α , respectively. As we know, the virtual electron (VE) progresses of occupied and unoccupied states are the main contribution to the overall SHG effect.⁸⁶

Meanwhile, we have also added the references ‘67. Godby R, Schlüter M, Sham L. Accurate Exchange-Correlation Potential for Silicon and Its Discontinuity on Addition of an Electron. *Phys. Rev. Lett.* **56**, 2415-2418 (1986).; 77. Clark S, *et al.* First principles methods using CASTEP. *Z Krist. Cryst. Mater.* **220**, 567-570 (2005).; 78. Perdew J, Burke K, Ernzerhof M. Generalized Gradient Approximation Made Simple. *Phys. Rev. Lett.* **77**, 3865-3868 (1996).; 79. Lin J, Qteish A, Payne M, Heine V. Optimized and transferable nonlocal separable ab initio pseudopotentials. *Phys. Rev. B* **47**, 4174-4180 (1993).; 80. Pickett W, Erwin S, Ethridge E. Reformulation of the LDA+U method for a local-orbital basis. *Phys. Rev. B* **58**, 1201-1209 (1998).; 81. German E, Faccio R, Mombrú A. A DFT+U study on structural, electronic, vibrational and thermodynamic

properties of TiO₂ polymorphs and hydrogen titanate: tuning the Hubbard ‘U-term’. *Commun. Phys.* **1**, 055006 (2017).; 82. Jiao Z, *et al.* Heteroanionic LaBrVIO₄ (VI = Mo, W): Excellence in Both Nonlinear Optical Properties and Photoluminescent Properties. *Chem. Mater.* **35**, 6998-7010 (2023).; 83. Aversa C, Sipe J. Nonlinear optical susceptibilities of semiconductors: Results with a length-gauge analysis. *Phys. Rev. B* **52**, 14636-14645 (1995).; 84. Lin J, Lee M, Liu Z, Chen C, Pickard C. Mechanism for linear and nonlinear optical effects in β -BaB₂O₄ crystals. *Phys. Rev. B* **60**, 13380-13389 (1999).; 85. Monkhorst H, Pack J. Special points for Brillouin-zone integrations. *Phys. Rev. B* **13**, 5188-5192 (1976).; 86. He R, Lin Z, Lee M, Chen C. Ab initio studies on the mechanism for linear and nonlinear optical effects in YAl₃(BO₃)₄. *J. Appl. Phys.* **109**, 103510 (2011).’ as References 67, 77-86.

3. LaMg₆Ga₆S₁₆ can exhibit the good NLO properties. This should also be mentioned in the abstract.

Response: Thanks for the reviewer’s kind comments and suggestions! According to the reviewer’s suggestion, the following sentences: ‘Furthermore, as LaMg₆Ga₆S₁₆ crystallizes in the non-centrosymmetric space group, *P*-6, it is the second-harmonic generation (SHG) active, possessing a comparable SHG response with classical AgGaS₂. In consideration of its wider band gap ($E_g = 3.0$ eV) and higher laser-induced damage threshold ($5\times$ AgGaS₂), LaMg₆Ga₆S₁₆ is also a promising nonlinear optical material’ have been added to abstract in the paper.

4. It is unclear that the IR spectra were recorded using spectrometer with ATR. To avoid misunderstanding, the authors should make a more detailed description on the measurement of IR spectra in experimental section.

Response: Thanks for the reviewer’s kind comment. According to the reviewer’s suggestion, “with ATR” has been added to the “IR and Raman spectroscopy” in the article.

5. Some minor grammatical errors can also be found in the manuscript. They should be

corrected before publication.

Response: Thanks a lot for the reviewer's comments! According to the reviewer's suggestions, we have revised the grammatical errors in the manuscript. For example, the predicate verb "were" has been changed to "was" (Line 16 in the left column of page 11), and the words "spectra", "band gaps" and "crystals" have been changed to "spectrum" (Line 15 in the left column of page 11), "band gap" (Line 18 in the left column of page 11) and "crystal" (Line 14 in the left column of page 10), respectively.

Reviewer #4 (Remarks to the Author):

This manuscript reports on the first synthesis of a La-chalcogenide, $\text{LaMg}_6\text{Ga}_6\text{S}_{16}$, where La is divalent cation. As the authors noted, the divalent La chalcogenide is rare because trivalent La is very stable, and it is noteworthy that this compound is chemically stable at high temperatures. In addition, this compound showed photoluminescence and second harmonic generation.

1. From the viewpoint of solid state chemistry, the chemical stability of this divalent La chalcogenide would be very interesting. In addition, this compound can be synthesized by using conventional solid phase synthesis.

Response: Thanks for the reviewer's good comments!

2. The optical properties are good but not excellent. In addition, the role of divalent La in the optical properties is unknown.

Response: Thanks for the reviewer's good comments! Actually, $\text{LaMg}_6\text{Ga}_6\text{S}_{16}$ has exhibited excellent IR NLO properties and interesting ultrabroad-band green emission. According to your suggestion, we have added more discussion on the excellent IR NLO properties and interesting ultrabroad-band green emission in the revised paper.

1) $\text{LaMg}_6\text{Ga}_6\text{S}_{16}$ can exhibit excellent IR NLO properties.

Generally, an excellent IR NLO crystal must satisfy the following structural and

properties requirements. Structurally, the material must crystallize in the noncentrosymmetric (NCS) space groups, which is the prerequisite for materials exhibit the SHG response. As for the properties, an excellent IR NLO crystal need to possess i) a large SHG responses ($d_{ij} > 0.5 \text{ AgGaS}_2$); ii) wide band gap ($E_g > 3.0 \text{ eV}$); iii) broad IR transmission covering two important atmospheric windows 3~5 μm and 8~12 μm ; iv) a moderate birefringence ($0.03 > \Delta n > 0.1$) to meet the phase-matching condition; v) ease for the crystal growth, and vi) good thermal and environmental stability (*Angew. Chem. Int. Ed.* **59**, 7514-7520 (2020); *Adv. Funct. Mater.* **32**, 2200231 (2022); *Coord. Chem. Rev.* **470**, 214706 (2022)). **Clearly, $\text{LaMg}_6\text{Ga}_6\text{S}_{16}$ crystallizes in the NCS structure and well satisfy the above properties**, including i) a large SHG response ($0.8 \times \text{AgGaS}_2$), ii) wide band gap ($E_g = 3.0 \text{ eV}$), iii) broad IR transmission covering two important atmospheric windows, 3~5 μm and 8~12 μm , iv) a moderate birefringence ($\Delta n = 0.04$), v) ease for crystal growth because of the congruently melting property, and vi) stable physicochemical properties.

The excellent NLO properties of $\text{LaMg}_6\text{Ga}_6\text{S}_{16}$ can be attributed to the particular contribution of La(II) cations. For the design of an IR NLO crystal, the greatest challenge is to balance the contradictory relationship between large SHG response and wide band gap. For example, for the commercialized IR NLO crystals including AgGaS_2 , AgGaSe_2 , and ZnGeP_2 , although they can exhibit large SHG responses, the small electronegativity difference between transition-cations (Ag^+ : 1.93, Zn^{2+} : 1.65) and anions (S^{2-} : 2.58, Se: 2.55, P: 2.19) makes them exhibit smaller band gaps ($< 3.0 \text{ eV}$). The small band gaps further result in their strong two-photon absorption and low laser-induced damage thresholds, and thus limit their applications in high-powder laser output. In order to increase the band gaps, the transition-cations have been changed into the alkali or alkaline-earth cations and a series of alkali or alkaline-earth chalcogenide NLO crystals with wide band gaps have been reported recently, such as LiGaS_2 (E_g : 4.15 eV), $\text{Na}_2\text{BaGeS}_4$ (E_g : 3.70 eV), $\text{Ba}_6\text{Zn}_7\text{Ga}_2\text{S}_{16}$ (E_g : 3.50 eV). But owing to the weak polarizability of alkali and alkaline-earth cations, the alkali and alkaline-earth chalcogenides can often exhibit reduced SHG responses, for example, LiGaS_2 ($0.7 \times \text{AgGaS}_2$), $\text{Na}_2\text{BaGeS}_4$ ($0.3 \times \text{AgGaS}_2$), $\text{Ba}_6\text{Zn}_7\text{Ga}_2\text{S}_{16}$ ($0.5 \times \text{AgGaS}_2$). That is

unfavorable for achieve the high laser conversion efficiency. But for rare-earth La^{2+} cations, it can exhibit not only the similar polarizability with Ag^+ or Zn^{2+} cations, but also comparable electronegativity (La^{2+} : 1.1) with alkali or alkaline-earth cations (0.8~1.0). So, it can achieve the better balance among the core properties. For the more detailed contribution of d electron of La (II) cations for the SHG response and band gaps, please see our response for Question 3 of Reviewer #2 and Question 2 of Reviewer #3.

2) ***LaMg₆Ga₆S₁₆ can show the interesting ultrabroad-band green emission.***

The materials with the ultrabroad emission are of great importance for diverse applications, including white light-emitting diodes (LED), 3D sensing, food analyzing, and other specific fields. Comparing the FWHM of emission spectrum of $\text{LaMg}_6\text{Ga}_6\text{S}_{16}$ with those in other well-developed rare earth-doped phosphors, including $\text{CaY}_2\text{HfAl}_4\text{O}_{12}:\text{Ce}^{3+}$ (FWHM: 120 nm), $\beta\text{-SiAlON}:\text{Yb}^{2+}$ (FWHM: 66 nm), $\text{Li}_2\text{SrSiO}_4:\text{Pr}^{3+}$ (FWHM: about 50 nm), $\beta\text{-SiAlON}:\text{Eu}^{2+}$ (FWHM: 55 nm), $\text{Ca}_3\text{SiO}_4\text{Cl}_2:\text{Eu}^{2+}$ (FWHM: 59 nm), $\text{Ba}_2\text{CaZn}_2\text{Si}_6\text{O}_{17}:\text{Eu}^{2+}$ (FWHM: 80 nm), $\text{Ba}_3\text{Si}_6\text{O}_{12}\text{N}_2:\text{Eu}^{2+}$ (FWHM: 75 nm), and $\text{Ba}_2\text{SiO}_4:\text{Eu}^{2+}$ (FWHM: 80 nm), it can be found that such an ultrabroad emission band covers almost the whole visible light region and could find applications in the field of warm-white LED lighting. *The ultrabroad emission of $\text{LaMg}_6\text{Ga}_6\text{S}_{16}$ can also be attributed to the particular electronic transition from upper-lying e_g to lower-lying t_{2g} of divalent La 5d orbitals in the low-coordinated octahedral crystal field (Figure 4e).* Please see more discussion for our response to Question 3 of Reviewer 1.

From the above discussion, we can see that $\text{LaMg}_6\text{Ga}_6\text{S}_{16}$ represents the first La(II)-based compounds with ***excellent IR NLO properties*** and ***interesting ultrabroad-band green emission*** because of the particular electronic characteristics of La^{2+} .

3. Molar percentage of La in this compound is very small, only 3%. Such small percentage would allow the anomalous valence of La, but the influence of divalent La on various properties of this compound might be very small. For example, in case of divalent Eu chalcogenides, EuO , EuS , etc, molar percentage of Eu is 50%, and the

sufficiently dense Eu ion governs their eminent magnetism in these compounds. To conclude, this study is very important in the fields of solid state chemistry, but will lack general readership, thus is not suitable for publication in this journal.

Response: Thanks for reviewer's comments! It is true that $\text{LaMg}_6\text{Ga}_6\text{S}_{16}$ has a low molar proportion of La in the structure and the low small percentage won't make an important contribution for the magnetism. But for the optical materials, the percentage will be enough to produce a decisive effect on optical or NLO properties. As mentioned in our response for your question #2, since rare-earth La^{2+} cation can exhibit similar polarizability with the transition Ag^+ and Zn^{2+} cations and comparable electropositivity with the alkali and alkaline-earth cations, *LaMg₆Ga₆S₁₆ can combine the advantages of large SHG responses of transition-cations chalcogenides and large band gaps of alkali and alkaline-earth chalcogenides and achieve a better balance between large SHG response and wide band gap, which is the main challenge for the design of IR NLO crystals. So, it indeed exhibits better NLO properties than commercialized AgGaS₂, which has been illustrated in Figure 5e.* In addition, because of the particular electronic characteristics of $\text{La}^{2+}(5d^1)$ in the low-coordinated octahedral crystal field, *LaMg₆Ga₆S₁₆ also displays an ultrabroad-band green emission at 500 nm with the FWHM of 127 nm, which almost cover the whole visible light region. That is rare observed in other luminescent materials.* That can provide some new insights for design of ultrabroad-band emission optical materials.

In summary, $\text{LaMg}_6\text{Ga}_6\text{S}_{16}$ is the first divalent La chalcogenide that can be synthesized by using conventional solid phase reaction. Also, because of the particular electronic characteristics of $\text{La}^{2+}(5d^1)$, $\text{LaMg}_6\text{Ga}_6\text{S}_{16}$ exhibits the excellent NLO properties and interesting ultrabroad-band green emission with the FWHM of 127 nm. *In our revised paper, according to your suggestion, we have added more discussion on its **excellent NLO properties** (please see Line 11 in the left column of page 8 in the paper) and **interesting luminescent properties** (please see Line 11 in the right column of page 6 and Line 13 in the right column of page 7 in the paper). We believe the revised paper will be suitable for Nature Communications. Thanks again for your all kind suggestions!*

REVIEWER COMMENTS

Reviewer #1 (Remarks to the Author):

The authors proposed that they synthesized the first stable crystalline La(II)-chalcogenide, LaMg₆Ga₆S₁₆. The luminescent properties and SHG response have been reported. The authors have responded the previous concerns and revised the manuscript accordingly. However, I still have serious concerns on the correct results of the existence of La(II), and the importance of this manuscript for the publication in Nature Communications.

1. I cannot find any references on the existence of La(II) in the solid-state compounds. What is the special character of this kind of metal chalcogenide. What is the driving force of the thermodynamics process for the existence of La(II). As I have mentioned previously, it is not reasonable to give the chemical formula of LaMg₆Ga₆S₁₆ and the chemical valence of La(II). If La(II)-chalcogenides are stable, there should be more examples.

2. The broad-band green emission at 500 nm is not a important findings in this work and the performance is poor for the photonics applications. The analysis in Figure 4d is not reasonable and useful for the emission mechanism. The reviewers wonder what is the origin of the emission positions at 500 nm. Can we modify the luminescence depending on the chemical compositions modification?

3. For the SHG applications, one expects to grow the big size crystals. The reviewers wonder there are some advantages for the La(II)-chalcogenide, LaMg₆Ga₆S₁₆.

Reviewer #2 (Remarks to the Author):

The authors have addressed all my concerns. The manuscript can be published in current version.

Reviewer #3 (Remarks to the Author):

The authors' responses and revision are satisfactory. I agree to accept this manuscript published on Nature Communications.

Reviewer #4 (Remarks to the Author):

This compound is unconventionally stable with the existence of divalent La ion, and shows similar luminescent properties to other existing materials. Thus, this study is interesting in fields of inorganic chemistry and/or optical materials. However, this compound is not superior to the existing luminescent materials. In addition, it is difficult to find remarkable multifunctional properties, although various physical properties are demonstrated in this manuscript. Accordingly, more specialized journal is suitable.

RESPONSE TO REVIEWERS' COMMENTS

Reviewer #1:

The authors proposed that they synthesized the first stable crystalline La(II)-chalcogenide, LaMg₆Ga₆S₁₆. The luminescent properties and SHG response have been reported. The authors have responded the previous concerns and revised the manuscript accordingly. However, I still have serious concerns on the correct results of the existence of La(II), and the importance of this manuscript for the publication in Nature Communications.

1. I cannot find any references on the existence of La(II) in the solid-state compounds. What is the special character of this kind of metal chalcogenide. What is the driving force of the thermodynamics process for the existence of La(II). As I have mentioned previously, it is not reasonable to give the chemical formula of LaMg₆Ga₆S₁₆ and the chemical valence of La(II). If La(II)-chalcogenides are stable, there should be more examples.

Response: Thanks for the reviewer's comments! Actually, there are several solid-state compounds with divalent La(II) atoms, including two metastable inorganic compounds LaO (*J. Solid State Chem.* **36**, 261-270 (1981); *Phys. Rev. B* **105**, L020508 (2022)) and LaS (*Phys. Status Solidi B Basic Res.* **174**, 435-447 (1992)), as well as two organic complexes, [K(18-crown-6)(OEt₂)][(C₅H₃(SiMe₃)_{2-1,3})₃La] (*Angew. Chem. Int. Ed.* **47**, 1488-1491 (2008)) and [K([2.2.2]crypt)][LaCp^{''}3](Cp^{''}=1,3-(SiMe₃)₂C₅H₃), [2.2.2]crypt=4,7,13,16,21,24-hexaoxa-1,10-diazabicyclo[8.8.8]hexacosane) (*J. Am. Chem. Soc.* **133**, 15914-15917 (2011)). Noted that in all the lanthanum monoxides and monochalcogenides, the divalent lanthanum (La²⁺) cations are coordinated by six Q (Q = O or S) atoms to form the [LaQ₆] octahedra. On the contrary, the high-oxidation-state lanthanum (La³⁺) is typically found with the higher-coordinated [LaQ_x] (x = 7 or 8) polyhedra, as observed in La₂S₃, LaGaS₃, La₂Ga₂GeS₈, La₆MgGe₂S₁₄, K₃LaP₂S₈, Ba₃La₄O₄(BO₃)₃X (X=F, Cl, Br). Therefore, the stable octahedral coordinated environments of lanthanum can be seen as the primary driving force for the formation of divalent lanthanum compounds.

For $\text{LaMg}_6\text{Ga}_6\text{S}_{16}$, its $[\text{Mg}_6\text{Ga}_6\text{S}_{16}]_\infty$ anionic framework is composed of the $[\text{MgS}_6]$ octahedra and $[\text{GaS}_4]$ tetrahedra through *edge-* and *face-sharing* connections. The strong *edge-* and *face-sharing* connections make the $[\text{Mg}_6\text{Ga}_6\text{S}_{16}]_\infty$ anionic framework very stable. *More importantly, the stable $[\text{Mg}/\text{Ga}-\text{S}]_\infty$ framework contains the C_{3h} symmetry along the c-axis. When cations with the suitable sizes filled in the voids, they can manifest the octahedral environments, such as Li^+ and Na^+ in $\text{M}_2\text{Mg}_6\text{Ga}_6\text{S}_{16}$ ($\text{M} = \text{Li}$ and Na) (*J. Am. Chem. Soc.* **144**, 21916-21925 (2022), and Ca in $\text{CaMg}_6\text{Ga}_6\text{S}_{16}$ (*Adv. Opt. Mater.* **11**, 2202147 (2022)). Therefore, when La cations are filled into the $[\text{Mg}_6\text{Ga}_6\text{S}_{16}]_\infty$ anionic framework, they also exhibit the octahedral environments and the chemical valence of +2. Therefore, we think that the stable $[\text{Mg}_6\text{Ga}_6\text{S}_{16}]_\infty$ anionic framework with octahedral voids filled by the $\text{La}(\text{II})$ cations are main the driving force for the existence of $\text{La}(\text{II})$. These have been illustrated by the paragraph at Line 8 in the right column of page 5 to Line 19 in the left column of page 6 in the manuscript.*

In addition, the nature of the divalent La^{2+} cations have also been identified by X-ray photoelectron spectroscopy, X-ray absorption near-edge structure and electron paramagnetic resonance as well as the photoluminescence spectron (Trivalent La^{3+} cations have no photoluminescence property). Therefore, we think the divalent La^{2+} cations in $\text{LaMg}_6\text{Ga}_6\text{S}_{16}$ are reasonable.

2. The broad-band green emission at 500 nm is not a important findings in this work and the performance is poor for the photonics applications. The analysis in Figure 4d is not reasonable and useful for the emission mechanism. The reviewers wonder what is the origin of the emission positions at 500 nm. Can we modify the luminescence depending on the chemical compositions modification?

Response: Thanks for reviewer's comments!

In order to study the origin of the luminescent property of $\text{LaMg}_6\text{Ga}_6\text{S}_{16}$, we further carried out the following measurements and analyses.

1) We measured the luminescent property of $\text{CaMg}_6\text{Ga}_6\text{S}_{16}$, $\text{SrMg}_6\text{Ga}_6\text{S}_{16}$ and $\text{LaMg}_6\text{Ga}_6\text{S}_{16}$, which all have the same $[\text{Mg}_6\text{Ga}_6\text{S}_{16}]_\infty$ anionic frameworks but the different cations. Experimental results indicate an obvious green emission at 500

nm can be observed for LaMg₆Ga₆S₁₆. But for CaMg₆Ga₆S₁₆ and SrMg₆Ga₆S₁₆, they have no any photoluminescence property (Figure 4e, Figure S6 and Figure S7). These suggest that the luminescent properties of LaMg₆Ga₆S₁₆ should originate from La cations, rather than the [Mg₆Ga₆S₁₆]_∞ anionic frameworks.

- 2) For La elements, it is well-known that the trivalent La³⁺ cations have no photoluminescence property (*Chem. Mater.* **21**, 1955-1961 (2009)). We also measured the photoluminescence property of La₂S₃ with the trivalent La³⁺ cations (Figure S8). It indeed has no any emissions. These further indicate the nature of the divalent La²⁺ (5d¹) cations in LaMg₆Ga₆S₁₆.
- 3) For La²⁺ (5d¹) cations, *Li, et al.* have calculated the octahedral crystal-field splitting gap through the first-principles calculations (*Phys. Rev. B* **105**, L020508 (2022)), which shows the octahedral crystal-field splitting gap, the upper-lying e_g and lower-lying t_{2g} is approximately 2.50 eV, which is consistent with the emission positions at 500 nm.

Based on the above measurements and analyses, it is clear that the emission positions at 500 nm of LaMg₆Ga₆S₁₆ should originate from the d-d transition of the La²⁺ within the low-coordinated octahedral crystal field. As observed in CaMg₆Ga₆S₁₆, SrMg₆Ga₆S₁₆ and LaMg₆Ga₆S₁₆, the chemical composition regulations do have the significant impact on luminescence properties (LaMg₆Ga₆S₁₆ can exhibit the photoluminescence property, but CaMg₆Ga₆S₁₆ and SrMg₆Ga₆S₁₆ cannot).

In a word, the divalent lanthanide inorganic compounds can exhibit unique electronic configurations and physicochemical properties. They are important for the many frontier fields such as superconductivity, magnetics, photoluminescence. But the synthesis of divalent lanthanum inorganic compounds remains a great challenge. LaMg₆Ga₆S₁₆ is the first La(II)-based compounds to exhibit luminescent properties. More importantly, this work can provide some new insights on how to design and synthesize the stable divalent lanthanum inorganic compounds. So, we think this work will be important and worthwhile to publish on Nature Communications.

Accordingly, the paragraph at Line 11 in the left column of the Page 6 has been

changed as following: ‘Based on these studies, the strong green emission observed in LaMg₆Ga₆S₁₆ does not stem from intrinsic defects of exceedingly low content. In order to find out the origin of PL property of LaMg₆Ga₆S₁₆, we also measured the luminescence features of CaMg₆Ga₆S₁₆ and SrMg₆Ga₆S₁₆, which are isomorphous to LaMg₆Ga₆S₁₆ with chemical substitutions from La to Ca or Sr. Experimental results indicate CaMg₆Ga₆S₁₆ (Figure 4d) and SrMg₆Ga₆S₁₆ (Figure S6) have no PL emission. Also, when the polycrystalline samples of CaMg₆Ga₆S₁₆, SrMg₆Ga₆S₁₆ and LaMg₆Ga₆S₁₆ were radiated by the UV irradiation, only LaMg₆Ga₆S₁₆ exhibited the green light emission (Figure S7). These results suggest that the PL property of LaMg₆Ga₆S₁₆ should come from the La cations, rather than the [Mg/Ga-S]_∞ anionic frameworks. But, the previous research⁴⁷ has confirmed that the trivalent La³⁺ cations cannot exhibit luminescent properties (we also measured the PL spectrum of La₂S₃ (Figure S8), which show that La₂S₃ with the trivalent La³⁺ cations have no PL property). So, these results also further indicate the nature of the divalent La²⁺ (5d¹) cations in LaMg₆Ga₆S₁₆. Referencing *Li’s, et al.* first-principles calculations on the octahedral crystal-field splitting gap between the upper-lying *e_g* and lower-lying *t_{2g}* for the La²⁺ 5d orbitals in monoxide LaO, we can conclude that the green emission position at 500 nm in LaMg₆Ga₆S₁₆ should originate from the *d-d* transition of the La²⁺ within the low-coordinated octahedral crystal field, because the octahedral crystal-field splitting gap for the La²⁺ 5d orbitals is approximately 2.50 eV (Figure 4e),^{1, 12, 25, 48} which is precisely consistent with the green emission position at 500 nm in LaMg₆Ga₆S₁₆.’ And three new Figures as Figure 4d, Figure S6 and Figure S7 have been added in the paper.

Figure 4d. Excitation-dependent PL spectra of $\text{CaMg}_6\text{Ga}_6\text{S}_{16}$ at room temperature.

Figure S6. Excitation-dependent PL spectra of $\text{SrMg}_6\text{Ga}_6\text{S}_{16}$ at room temperature.

Figure S7. Optical images of $\text{LaMg}_6\text{Ga}_6\text{S}_{16}$, $\text{CaMg}_6\text{Ga}_6\text{S}_{16}$, and $\text{SrMg}_6\text{Ga}_6\text{S}_{16}$ under UV irradiation at room temperature.

In addition, the new reference ‘47. Roof I, Jagau T, Wolfgang G. Zeier W, Smith M, Loye H. Crystal Growth of a New Series of Complex Niobates, LnKNaNbO_5 ($\text{Ln} = \text{La, Pr, Nd, Sm, Eu, Gd, and Tb}$): Structural Properties and Photoluminescence. *Chem. Mater.* **21**, 1955-1961 (2009).’ was also added as Reference 47.

3. For the SHG applications, one expects to grow the big size crystals. The reviewers wonder there are some advantages for the La(II)-chalcogenide, $\text{LaMg}_6\text{Ga}_6\text{S}_{16}$.

Response: Thanks for the reviewer’s comments! Indeed, the large-size and high-quality crystal growth is important for the practical application of a NLO crystal. But obviously, the growth of chalcogenide crystals needs a sealed environment, and their crystal growth is generally very time-consuming. From the discovery of a new NLO crystal to its large-size crystal growth, it will take several years or longer time and contain many efforts from different fields, such as chemistry, material science, and optics, etc. Of course, we are also working on growing the large-size $\text{LaMg}_6\text{Ga}_6\text{S}_{16}$ crystals.

For the SHG applications, the advantage $\text{LaMg}_6\text{Ga}_6\text{S}_{16}$ is also obvious.

- 1) **$\text{LaMg}_6\text{Ga}_6\text{S}_{16}$ has the larger NLO coefficients.** Clearly, compared with the reported $\text{AeMg}_6\text{Ga}_6\text{S}_{16}$ ($\text{Ae} = \text{Ca, Sr, Ba}$), the calculated SHG coefficients of $\text{LaMg}_6\text{Ga}_6\text{S}_{16}$ ($d_{11} = 12.27 \text{ pm/V}$ and $d_{22} = 4.00 \text{ pm/V}$) are greater than that of the $\text{AeMg}_6\text{Ga}_6\text{S}_{16}$ ($\text{Ae} = \text{Ca, Sr, Ba}$) (Table S6), suggesting divalent La make the positive contribution to NLO response of $\text{LaMg}_6\text{Ga}_6\text{S}_{16}$. These have been illustrated by the paragraph at Line 19 to Line 26 in the left column of page 6 in the manuscript.
- 2) **$\text{LaMg}_6\text{Ga}_6\text{S}_{16}$ may exhibit the better crystal growth behavior.** Impressively, compared with alkali and alkaline-earth cations, rare-earth La^{2+} cation has more complex electronic configurations and a bigger atomic radius. Therefore, it is polarizable and La-S bonds can exhibit decreased ionicity, which is an advantageous factor for the growth of large-size $\text{LaMg}_6\text{Ga}_6\text{S}_{16}$ crystals (*Chem. Mater.* **33**, 4240–4246 (2021); *Chem. Mater.* **34**, 8004–8012 (2022); *Opt. Express* **18**, 237–243 (2010)).

3) *LaMg₆Ga₆S₁₆ can satisfy all rigid structural and properties' requirement for IR NLO crystals.* Generally, an excellent IR NLO crystal must satisfy the following structural and properties requirements. Structurally, the material must crystallize in the noncentrosymmetric (NCS) space groups, which is the prerequisite for materials exhibit the SHG response. As for the properties, an excellent IR NLO crystal need to possess i) a large SHG responses ($d_{ij} > 0.5 \text{ AgGaS}_2$); ii) wide band gap ($E_g > 3.0 \text{ eV}$); iii) broad IR transmission covering two important atmospheric windows 3~5 μm and 8~12 μm ; iv) a moderate birefringence ($0.03 > \Delta n > 0.1$) to meet the phase-matching condition, and v) good thermal and environmental stability (*Angew. Chem. Int. Ed.* **59**, 7514-7520 (2020); *Adv. Funct. Mater.* **32**, 2200231 (2022); *Coord. Chem. Rev.* **470**, 214706 (2022)). Clearly, *LaMg₆Ga₆S₁₆ crystallizes in the NCS structure and well satisfy the above properties, including i) a large SHG response ($0.8 \times \text{AgGaS}_2$), ii) wide band gap ($E_g = 3.0 \text{ eV}$), iii) broad IR transmission covering two important atmospheric windows, 3~5 μm and 8~12 μm , iv) a moderate birefringence ($\Delta n = 0.04$), v) stable physicochemical properties.*

These indicate that the La(II)-chalcogenide, *LaMg₆Ga₆S₁₆* does have some advantages for the SHG applications.

Reviewer #4 (Remarks to the Author):

This compound is unconventionally stable with the existence of divalent La ion, and shows similar luminescent properties to other existing materials. Thus, this study is interesting in fields of inorganic chemistry and/or optical materials. However, this compound is not superior to the existing luminescent materials. In addition, it is difficult to find remarkable multifunctional properties, although various physical properties are demonstrated in this manuscript. Accordingly, more specialized journal is suitable.

Response: Thanks for the reviewer's comments! Although *LaMg₆Ga₆S₁₆* does not exhibit impressive luminescent properties, as you highlighted, *it represents the first chemically stable La(II)-based chalcogenide that can be synthesized by using*

conventional solid phase reaction. This signifies a noteworthy advancement in the chemistry of lanthanides. More importantly, because of the particular electronic characteristics of $\text{La}^{2+}(5d^1)$, $\text{LaMg}_6\text{Ga}_6\text{S}_{16}$ displays an ultrabroad-band green emission at 500 nm. *Clearly, this is also the first time for La(II)-based compounds to exhibit luminescent properties. That can provide some new insights for the design of luminescent materials centered around lanthanum.* In addition, $\text{LaMg}_6\text{Ga}_6\text{S}_{16}$ crystallizes in the non-centrosymmetric structure and exhibits excellent nonlinear optical properties, including i) a large SHG response ($0.8 \times \text{AgGaS}_2$), ii) wide band gap ($E_g = 3.0$ eV), iii) broad IR transmission covering two important atmospheric windows, 3~5 μm and 8~12 μm , iv) a moderate birefringence ($\Delta n = 0.04$), v) ease for crystal growth because of the congruently melting property, and vi) stable physicochemical properties, *which is better than commercialized AgGaS_2 (Figure 5e) and a promising nonlinear optical material.* We believe this intriguing research will be suitable for *Nature Communications*. Thanks again for your comments!

Thank you for the opportunity to publish in *Nature Communications*.

Sincerely yours,
Hongwei Yu
Professor of Chemistry
Institute of Functional Crystal
Tianjin University of Technology
No. 391 Bin Shui Xi Dao Road
Tianjin 300384, China
Phone: (086)022-60216056
E-mail: yuhw@email.tjut.edu.cn

REVIEWERS' COMMENTS

Reviewer #1 (Remarks to the Author):

The authors have responded my previous comments. The authors proposed the viewpoint that "the stable $[\text{Mg}_6\text{Ga}_6\text{S}_{16}]_\infty$ anionic framework with octahedral voids filled by the La(II) cations are the driving force for the existence of La(II)". If this is right, can the authors prepare other similar divalent lanthanide inorganic compounds, not only the $\text{LaMg}_6\text{Ga}_6\text{S}_{16}$ in this work. Other additional work is needed to support this explanation.

The luminescence properties are really very poor, and this kind of compounds cannot say the good light emitting materials.

Based on the two serious concerns, I still don't support the publication of this manuscript in Nature Communications.

RESPONSE TO REVIEWERS' COMMENTS

Reviewer #1 (Remarks to the Author):

The authors have responded my previous comments. The authors proposed the viewpoint that "the stable $[\text{Mg}_6\text{Ga}_6\text{S}_{16}]_\infty$ anionic framework with octahedral voids filled by the La(II) cations are the driving force for the existence of La(II)". If this is right, can the authors prepare other similar divalent lanthanide inorganic compounds, not only the $\text{LaMg}_6\text{Ga}_6\text{S}_{16}$ in this work. Other additonal work is needed to support this explanation. The luminescence properties are really very poor, and this kind of compounds cannot say the good light emitting materials. Based on the two serious concerns, I still don't support the publication of this manuscript in Nature Communications.

Response: Thanks for the reviewer's comments!

As described in our previous response to question of reviewers' comments, all the divalent lanthanum (La^{2+}) cations are seen in the octahedral environments, while the trivalent lanthanum (La^{3+}) cations are typically found with the higher-coordinated $[\text{LaQ}_x]$ ($x = 7$ or 8) polyhedra. Therefore, the stable octahedral coordinated environments of lanthanum can be seen as the primary driving force for the formation of divalent lanthanum compounds. In $\text{LaMg}_6\text{Ga}_6\text{S}_{16}$, through using MgS_6 octahedra coupling tetrahedral single/double chains, a stable octahedral environment was constructed. That allows the existence of La(II) cations and the nature of the divalent La^{2+} cations has also been well-identified by XPS, XANES, EPR and PL. However, it is also clear the different divalent lanthanide ions have the different coordination states. So, other divalent lanthanide compounds cannot always be synthesized with the same condition. Therefore, to synthesize more divalent lanthanide inorganic compounds, more attempts will be needed. We will try them in our future research work.

As for the properties, $\text{LaMg}_6\text{Ga}_6\text{S}_{16}$ has exhibited the excellent NLO property and is potential as a NLO crystal. For luminescence properties, although $\text{LaMg}_6\text{Ga}_6\text{S}_{16}$ does not exhibit very exciting luminescence properties, it is the first chemically stable La(II)-based chalcogenide and represents the first example for La(II)-based compounds to exhibit luminescent properties. Owing to the electronic characteristics of $\text{La}^{2+}(5d^1)$,

$\text{LaMg}_6\text{Ga}_6\text{S}_{16}$ also displays an ultrabroad-band green emission at 500 nm. All of these will be interesting for exploring new stable low-valent lanthanide compounds.